# Provable Zero-Shot Generalization in Offline Reinforcement Learning

Zhiyong Wang [1]   Chen Yang [2]   John C.S. Lui [1]   Dongruo Zhou [2]

## Abstract

In this work, we study offline reinforcement learning (RL) with zero-shot generalization property (ZSG), where the agent has access to an offline dataset including experiences from different environments, and the goal of the agent is to train a policy over the training environments which performs well on test environments without further interaction. Existing work showed that classical offline RL fails to generalize to new, unseen environments. We propose pessimistic empirical risk minimization (PERM) and pessimistic proximal policy optimization (PPPO), which leverage pessimistic policy evaluation to guide policy learning and enhance generalization. We show that both PERM and PPPO are capable of finding a near-optimal policy with ZSG. Our result serves as a first step in understanding the foundation of the generalization phenomenon in offline reinforcement learning.

## 1. Introduction

Offline reinforcement learning (RL) has become increasingly significant in modern RL because it eliminates the need for direct interaction between the agent and the environment; instead, it relies solely on learning from an offline training dataset. However, in practical applications, the offline training dataset often originates from a different environment than the one of interest. This discrepancy necessitates evaluating RL agents in a generalization setting, where the training involves a finite number of environments drawn from a specific distribution, and the testing is conducted on a distinct set of environments from the same or different distribution. This scenario is commonly referred to as the zero-shot generalization (ZSG) challenge which has

[1]Department of Computer Science and Engineering, The Chinese University of Hong Kong, Hong Kong SAR, China [2]Department of Computer Science, Indiana University Bloomington, Bloomington, USA. Correspondence to: Dongruo Zhou <dz13@iu.edu>.

*Proceedings of the $42^{nd}$ International Conference on Machine Learning*, Vancouver, Canada. PMLR 267, 2025. Copyright 2025 by the author(s).

been studied in online RL(Rajeswaran et al., 2017; Machado et al., 2018; Justesen et al., 2018; Packer et al., 2019; Zhang et al., 2018a;b), as the agent receives no training data from the environments it is tested on.

A number of recent empirical studies (Mediratta et al., 2023; Yang et al., 2023; Mazoure et al., 2022) have recognized this challenge and introduced various offline RL methodologies that are capable of ZSG. Notwithstanding the lack of theoretical backing, these methods are somewhat restrictive; for instance, some are only effective for environments that vary solely in observations(Mazoure et al., 2022), while others are confined to the realm of imitation learning(Yang et al., 2023), thus limiting their applicability to a comprehensive framework of offline RL with ZSG capabilities. Concurrently, theoretical advancements (Bose et al., 2024; Ishfaq et al., 2024) in this domain have explored multi-task offline RL by focusing on representation learning. These approaches endeavor to derive a low-rank representation of states and actions, which inherently requires additional interactions with the downstream tasks to effectively formulate policies based on these representations. Therefore, we raise a natural question:

*Can we design provable offline RL with zero-shot generalization ability?*

We propose novel offline RL frameworks that achieve ZSG to address this question affirmatively. Our contributions are listed as follows.

- We first analyze when existing offline RL approaches fail to generalize without further algorithm modifications. Specifically, we prove that if the offline dataset does not contain context information, then it is impossible for vanilla RL that equips a Markovian policy to achieve a ZSG property. We show that the offline dataset from a contextual Markov Decision Process (MDP) is not distinguishable from a vanilla MDP which is the average of contextual Markov Decision Process over all contexts. Such an analysis verifies the necessity of new RL methods with ZSG property.

- We propose two meta-algorithms called pessimistic empirical risk minimization (PERM) and pessimistic proximal policy optimization (PPPO) that enable ZSG for offline

RL (Jin et al., 2021). In detail, both of our algorithms take a pessimistic policy evaluation (PPE) oracle as its component and output policies based on offline datasets from multiple environments. Our result shows that the sub-optimalities of the output policies are bounded by both the supervised learning error, which is controlled by the number of different environments, and the reinforcement learning error, which is controlled by the coverage of the offline dataset to the optimal policy. Please refer to Table 1 for a summary of our results. To the best of our knowledge, our proposed algorithms are the first offline RL methods that provably enjoy the ZSG property.

**Notation** We use lower case letters to denote scalars, and use lower and upper case bold face letters to denote vectors and matrices respectively. We denote by $[n]$ the set $\{1, \ldots, n\}$. For a vector $\mathbf{x} \in \mathbb{R}^d$ and a positive semi-definite matrix $\mathbf{\Sigma} \in \mathbb{R}^{d \times d}$, we denote by $\|\mathbf{x}\|_2$ the vector's Euclidean norm and define $\|\mathbf{x}\|_{\mathbf{\Sigma}} = \sqrt{\mathbf{x}^\top \mathbf{\Sigma} \mathbf{x}}$. For two positive sequences $\{a_n\}$ and $\{b_n\}$ with $n = 1, 2, \ldots$, we write $a_n = O(b_n)$ if there exists an absolute constant $C > 0$ such that $a_n \leq Cb_n$ holds for all $n \geq 1$ and write $a_n = \Omega(b_n)$ if there exists an absolute constant $C > 0$ such that $a_n \geq Cb_n$ holds for all $n \geq 1$. We use $\widetilde{O}(\cdot)$ to further hide the polylogarithmic factors. We use $(x_i)_{i=1}^n$ to denote sequence $(x_1, ..., x_n)$, and we use $\{x_i\}_{i=1}^n$ to denote the set $\{x_1, ..., x_n\}$. We use $\text{KL}(p\|q)$ to denote the KL distance between distributions $p$ and $q$, defined as $\int p \log(p/q)$. We use $\mathbb{E}[x], \mathbb{V}[x]$ to denote expectation and variance of a random variable $x$.

The remaining parts are organized as follows. In Section 2, we discuss related works. In Section 3, we introduce the setting of our work. In Section 4, we analyze when existing offline RL approaches (Jin et al., 2021) fail to generalize without further algorithm modifications. In Section 5, we introduce our proposed meta-algorithms and provide their theoretical guarantees. In Section 6, we specify our meta-algorithms and analysis to a more concrete linear MDP setting. Finally, in Section 7, we conclude our work and propose some future directions.

## 2. Related works

**Offline RL** Offline reinforcement learning (RL) (Ernst et al., 2005; Riedmiller, 2005; Lange et al., 2012; Levine et al., 2020; Wang et al., 2024) addresses the challenge of learning a policy from a pre-collected dataset without direct online interactions with the environment. A central issue in offline RL is the inadequate dataset coverage, stemming from a lack of exploration (Levine et al., 2020; Liu et al., 2020). A common strategy to address this issue is the application of the pessimism principle, which penalizes the estimated value of under-covered state-action pairs. Numerous studies have integrated pessimism into various single-environment

offline RL methodologies. This includes model-based approaches (Rashidinejad et al., 2021; Uehara and Sun, 2021; Jin et al., 2021; Yu et al., 2020; Xie et al., 2021b; Uehara et al., 2021; Yin et al., 2022), model-free techniques (Kumar et al., 2020; Wu et al., 2021; Bai et al., 2022; Ghasemipour et al., 2022; Yan et al., 2023), and policy-based strategies (Rezaeifar et al., 2022; Xie et al., 2021a; Zanette et al., 2021; Nguyen-Tang and Arora, 2024). (Yarats et al., 2022) has observed that with sufficient offline data diversity and coverage, the need for pessimism to mitigate extrapolation errors and distribution shift might be reduced. To the best of our knowledge, we are the first to theoretically study the generalization ability of offline RL in the contextual MDP setting.

**Generalization in online RL** There are extensive empirical studies on training online RL agents that can generalize to new transition and reward functions (Rajeswaran et al., 2017; Machado et al., 2018; Justesen et al., 2018; Packer et al., 2019; Zhang et al., 2018a;b; Nichol et al., 2018; Cobbe et al., 2018; Küttler et al., 2020; Bengio et al., 2020; Bertran et al., 2020; Ghosh et al., 2021; Kirk et al., 2023; Juliani et al., 2019; Ajay et al., 2021; Samvelyan et al., 2021; Frans and Isola, 2022; Albrecht et al., 2022; Ehrenberg et al., 2022; Song et al., 2020; Lyle et al., 2022; Ye et al., 2020; Lee et al., 2020; Jiang et al.). They use techniques including implicit regularization (Song et al., 2020), data augmentation (Ye et al., 2020; Lee et al., 2020), uncertainty-driven exploration (Jiang et al.), successor feature (Touati et al., 2023), etc. These works focus mostly on the online RL setting and do not provide theoretical guarantees, thus differing a lot from ours. Moreover, (Touati et al., 2023) has studied zero-shot generalization in offline RL, but to unseen reward functions rather than unseen environments. Addtional related works that have studied zero-shot RL include (Park et al., 2024; Jeen et al., 2023).

There are also some recent works aimed at understanding online RL generalization from a theoretical perspective. Wang et al. (2019) examined a specific class of reparameterizable RL problems and derived generalization bounds using Rademacher complexity and the PAC-Bayes bound. Malik et al. (2021) established lower bounds and introduced efficient algorithms that ensure a near-optimal policy for deterministic MDPs. A recent work (Ye et al., 2023) studied how much pre-training can improve online RL test performance under different generalization settings. To the best of our knowledge, no previous work exists on theoretical understanding of the zero-shot generalization of offline RL.

Our paper is also related to recent works studying multi-task learning in reinforcement learning (RL) (Brunskill and Li, 2013; Tirinzoni et al., 2020; Hu et al., 2021; Zhang and Wang, 2021; Lu et al., 2021; Bose et al., 2024; Ishfaq et al., 2024; Zhang et al., 2023), which focus on transferring

*Table 1.* Summary of our algorithms and their suboptimality gaps, where $\mathcal{A}$ is the action space, $H$ is the length of episode, $n$ is the number of environments in the offline dataset. Note that in the multi-environment setting, $\pi^*$ is the near-optimal policy w.r.t. expectation (defined in Section 3). $\mathcal{N}$ is the covering number of the policy space $\Pi$ w.r.t. distance $\mathrm{d}(\pi^1, \pi^2) = \max_{s \in \mathcal{S}, h \in [H]} \|\pi_h^1(\cdot|s) - \pi_h^2(\cdot|s)\|_1$. The uncertainty quantifier $\Gamma_{i,h}$ are tailored with the oracle return in the corresponding algorithms (details are in Section 5).

| Algorithm | Suboptimality Gap |
|---|---|
| PERM (our Algo.2) | $\sqrt{\log(\mathcal{N})/n} + n^{-1} \sum_{i=1}^n \sum_{h=1}^H \mathbb{E}_{i,\pi^*}\left[\Gamma_{i,h}(s_h, a_h) \,\middle|\, s_1 = x_1\right]$ |
| PPPO (our Algo.3) | $\sqrt{\log|\mathcal{A}| H^2/n} + n^{-1} \sum_{i=1}^n \sum_{h=1}^H \mathbb{E}_{i,\pi^*}\left[\Gamma_{i,h}(s_h, a_h) \,\middle|\, s_1 = x_1\right]$ |

the knowledge learned from upstream tasks to downstream ones. Additionally, these works typically assume that all tasks share similar transition dynamics or common representations while we do not. Meanwhile, they typically require the agent to interact with the downstream tasks, which does not fall into the ZSG regime.

## 3. Preliminaries

**Contextual MDP** We study *contextual episodic MDPs*, where each MDP $\mathcal{M}_c$ is associated with a context $c \in C$ belongs to the context space $C$. Furthermore, $\mathcal{M}_c = \{M_{c,h}\}_{h=1}^H$ consists of $H$ different individual MDPs, where each individual MDP $M_{c,h} := (\mathcal{S}, \mathcal{A}, P_{c,h}(s'|s,a), r_{c,h}(s,a))$. Here $\mathcal{S}$ denotes the state space, $\mathcal{A}$ denotes the action space, $P_{c,h}$ denotes the transition function and $r_{c,h}$ denotes the reward function at stage $h$. We assume the starting state for each $\mathcal{M}_c$ is the same state $x_1$. In this work, we interchangeably use "environment" or MDP to denote the MDP $\mathcal{M}_c$ with different contexts.

**Policy and value function** We denote the policy $\pi_h$ at stage $h$ as a mapping $\mathcal{S} \to \Delta(\mathcal{A})$, which maps the current state to a distribution over the action space. We use $\pi = \{\pi_h\}_{h=1}^H$ to denote their collection. Then for any episodic MDP $\mathcal{M}$, we define the value function for some policy $\pi$ as

$$V_{\mathcal{M},h}^\pi(x) := \mathbb{E}[r_h + \ldots + r_H | s_h = x, a_{h'} \sim \pi_{h'},$$
$$r_{h'} \sim r_{h'}(s_{h'}, a_{h'}), s_{h'+1} \sim P_{h'}(\cdot|s_{h'}, a_{h'}), \ h' \geq h],$$
$$Q_{M,h}^\pi(x,a) := \mathbb{E}[r_h + \ldots + r_H | s_h = x, a_h = a,$$
$$r_h \sim r_h(s_h, a_h), s_{h'} \sim P_{h'-1}(\cdot|s_{h'-1}, a_{h'-1}), a_{h'} \sim \pi_{h'},$$
$$r_{h'} \sim r_{h'}(s_{h'}, a_{h'}), \ h' \geq h+1].$$

For any individual MDP $M$ with reward $r$ and transition dynamic $P$, we denote its Bellman operator $[\mathbb{B}_M f](x,a)$ as $[\mathbb{B}_M f](s,a) := \mathbb{E}[r_h(s,a) + f(s')|s' \sim P(\cdot|s,a)]$. Then we have the well-known Bellman equation

$$V_{\mathcal{M},h}^\pi(x)$$
$$= \langle Q_{\mathcal{M},h}^\pi(x,\cdot), \pi_h(\cdot|x)\rangle_{\mathcal{A}}, \ Q_{\mathcal{M},h}^\pi(x,a) = [\mathbb{B}_{M_h} V_{\mathcal{M},h+1}^\pi](x,a).$$

For simplicity, we use $V_{c,h}^\pi, Q_{c,h}^\pi, \mathbb{B}_{c,h}$ to denote $V_{\mathcal{M}_c,h}^\pi, Q_{\mathcal{M}_c,h}^\pi, \mathbb{B}_{M_{c,h}}$. We also use $\mathbb{P}_c$ to denote $\mathbb{P}_{\mathcal{M}_c}$, the joint distribution of any potential objects under the $\mathcal{M}_c$ episodic MDP. We would like to find the near-optimal policy $\pi^*$ w.r.t. expectation, i.e., $\pi^* := \mathrm{argmax}_{\pi \in \Pi} \mathbb{E}_{c \sim C} V_{c,1}^\pi(x_c)$, where $\Pi$ is the set of collection of Markovian policies, and with a little abuse of notation, we use $\mathbb{E}_{c \sim C}$ to denote the expectation taken w.r.t. the i.i.d.

sampling of context $c$ from the context space. Then our goal is to develop the *generalizable RL* with small *zero-shot generalization gap (ZSG gap)*, defined as follows:

$$\mathrm{SubOpt}(\pi) := \mathbb{E}_{c \sim C}\left[V_{c,1}^{\pi^*}(x_1)\right] - \mathbb{E}_{c \sim C}\left[V_{c,1}^\pi(x_1)\right].$$

**Remark 1** *We briefly compare generalizable RL with several related settings. Robust RL (Pinto et al., 2017) aims to find the best policy for the worst-case environment, whereas generalizable RL seeks a policy that performs well in the average-case environment. Meta-RL (Beck et al., 2023) enables few-shot adaptation to new environments, either through policy updates (Finn et al., 2017) or via history-dependent policies (Duan et al., 2016). In contrast, generalizable RL primarily focuses on the zero-shot setting. In the general POMDP framework (Cassandra et al., 1994), agents need to maintain history-dependent policies to implicitly infer environment information, while generalizable RL aims to discover a single state-dependent policy that generalizes well across all environments.*

**Remark 2** *Ye et al. (2023) showed that in online RL, for a certain family of contextual MDPs, it is inherently impossible to determine an optimal policy for each individual MDP. Given that offline RL poses greater challenges than its online counterpart, this impossibility extends to finding optimal policies for each MDP in a zero-shot offline RL setting as well, which justifies our optimization objective on the ZSG gap. Moreover, Ye et al. (2023) showed that the few-shot RL is able to find the optimal policy for individual MDPs. Clearly, such a setting is stronger than ours, and the additional interactions are often hard to be satisfied in real-world practice. We leave the study of such a setting for future work.*

**Offline RL data collection process** The data collection process is as follows. An experimenter i.i.d. samples number $n$ of contextual episodic MDP $M_i$ from the context set (*e.g.*, $i \sim C$). For each episodic MDP $M_i$, the experimenter collects dataset $\mathcal{D}_i := \{(x_{i,h}^\tau, a_{i,h}^\tau, r_{i,h}^\tau)_{h=1}^H\}_{\tau=1}^K$ which includes $K$ trajectories. Note that the action $a_{i,h}^\tau$ selected by the experimenter can be arbitrary, and it does not need to follow a specific behavior policy (Jin et al., 2021). We assume that $\mathcal{D}_i$ is compliant with the episodic MDP $\mathcal{M}_i$, which is defined as follows.

**Definition 3 ((Jin et al., 2021))** *For a dataset* $\mathcal{D}_i := \{(x_{i,h}^\tau, a_{i,h}^\tau, r_{i,h}^\tau)_{h=1}^H\}_{\tau=1}^K$, *let* $\mathbb{P}_{\mathcal{D}_i}$ *be the joint distribution of the data collecting process. We say* $\mathcal{D}_i$ *is compliant with episodic MDP* $\mathcal{M}_i$ *if for any* $x' \in \mathcal{S}, r', \tau \in [K], h \in [H]$, *we have*

$$\mathbb{P}_{\mathcal{D}_i}(r_{i,h}^\tau = r', x_{i,h+1}^\tau = x' | \{(x_{i,h}^j, a_{i,h}^j)\}_{j=1}^\tau,$$
$$\{(r_{i,h}^j, x_{i,h+1}^j)\}_{j=1}^{\tau-1})$$
$$= \mathbb{P}_i(r_{i,h}(s_h, a_h) = r', s_{h+1} = x' | s_h = x_h^\tau, a_h = a_h^\tau).$$

In general, we claim $\mathcal{D}_i$ is compliant with $\mathcal{M}_i$ when the conditional distribution of any tuple of reward and next state in $\mathcal{D}_i$ follows the conditional distribution determined by MDP $\mathcal{M}_i$.

# 4. Offline RL without context indicator information

In this section, we show that directly applying existing offline RL algorithms over datasets from multiple environments *without* maintaining their identity information cannot yield a sufficient ZSG property, which is aligned with the existing observation of the poor generalization performance of offline RL (Mediratta et al., 2023).

In detail, given contextual MDPs $\mathcal{M}_1, ..., \mathcal{M}_n$ and their corresponding offline datasets $\mathcal{D}_1, ..., \mathcal{D}_n$, we assume the agent only has the access to the offline dataset $\bar{\mathcal{D}} = \cup_{i=1}^n \mathcal{D}_i$, where $\bar{\mathcal{D}} = \{(x_{c_\tau, h}^\tau, a_{c_\tau, h}^\tau, r_{c_\tau, h}^\tau)_{h=1}^H\}_{\tau=1}^K$. Here $c_\tau \in C$ is the context information of trajectory $\tau$, which is *unknown* to the agent. To explain why offline RL without knowing context information performs worse, we have the following proposition suggesting the offline dataset from multiple MDPs is not distinguishable from an "average MDP" if the offline dataset does not contain context information.

**Proposition 4** $\bar{\mathcal{D}}$ *is compliant with* average MDP $\bar{\mathcal{M}} := \{\bar{M}_h\}_{h=1}^H$, $\bar{M}_h := (\mathcal{S}, \mathcal{A}, H, \bar{P}_h, \bar{r}_h)$,

$$\bar{P}_h(x'|x, a) := \mathbb{E}_{c \sim C} \frac{P_{c,h}(x'|x, a)\mu_{c,h}(x, a)}{\mathbb{E}_{c \sim C}\mu_{c,h}(x, a)},$$
$$\mathbb{P}(\bar{r}_h = r|x, a) := \mathbb{E}_{c \sim C} \frac{\mathbb{P}(\bar{r}_{c,h} = r|x, a)\mu_{c,h}(x, a)}{\mathbb{E}_{c \sim C}\mu_{c,h}(x, a)},$$

*where* $\mu_{c,h}(\cdot, \cdot)$ *is the data collection distribution of* $(s, a)$ *at stage* $h$ *in dataset* $\mathcal{D}_c$.

**Proof** See Appendix A.1. ∎

Proposition 4 suggests that if no context information is revealed, then the merged offline dataset $\bar{\mathcal{D}}$ is equivalent to a dataset collected from the average MDP $\bar{\mathcal{M}}$. Therefore, for any offline RL which outputs a Markovian policy, it converges to the optimal policy $\bar{\pi}^*$ of the average MDP $\bar{\mathcal{M}}$.

In general, $\bar{\pi}^*$ can be very different from $\pi^*$ when the transition probability functions of each environment are different. For example, consider the 2-context cMDP problem shown in Figure 1, each context consists of one state and three possible actions. The offline dataset distributions $\mu$ are marked on the arrows that both of the distributions are following near-optimal policy. By Proposition 4, in average MDP $\bar{\mathcal{M}}$ the reward of the middle action is deterministically 0, while both upper and lower actions are deterministically 1. As a result, the optimal policy $\bar{\pi}^*$ will only have positive probabilities toward upper and lower actions. This leads to $\mathbb{E}_{c \sim C}[V_{c,1}^{\bar{\pi}^*}(x_1)] = 0$, though we can see that $\pi^*$ is deterministically choosing the middle action and $\mathbb{E}_{c \sim C}[V_{c,1}^{\pi^*}(x_1)] = 0.5$. This theoretically illustrates that the generalization ability of offline RL algorithms without leveraging context information is weak. In sharp contrast, imitation learning such as behavior cloning (BC) converges to the teacher policy that is independent of the specific MDP. Therefore, offline RL methods such as CQL (Kumar et al., 2020) might enjoy worse generalization performance compared with BC, which aligns with the observation made by Mediratta et al. (2023).

*Figure 1.* Two Contextual MDPs with the same compliant average MDPs. The discrete contextual space is defined as $C = \{v, w\}$ and both MDPs satisfies $\mathcal{S} = \{x_1\}, \mathcal{A} = \{a_1, a_2, a_3\}, H = 1$. The data collection distributions $\mu$ and rewards $r$ for each action of each context are specified in the graph.

# 5. Provable offline RL with zero-shot generalization

In this section, we propose offline RL with small ZSG gaps. We show that two popular offline RL approaches, *model-based RL* and *policy optimization-based RL*, can output RL agent with ZSG ability, with a pessimism-style modification that encourages the agent to follow the offline dataset pattern.

## 5.1. Pessimistic policy evaluation

We consider a meta-algorithm to evaluate any policy $\pi$ given an offline dataset, which serves as a key component in our

---

**Algorithm 1** $\underline{P}$essimistic $\underline{P}$olicy $\underline{E}$valuation (PPE)

---

**Require:** Offline dataset $\{\mathcal{D}_{i,h}\}_{h=1}^H$, policy $\pi = (\pi_h)_{h=1}^H$, confidence probability $\delta \in (0,1)$.

1: Initialize $\widehat{V}_{i,H+1}^\pi(\cdot) \leftarrow 0, \ \forall i \in [n]$.
2: **for** step $h = H, H-1, \ldots, 1$ **do**
3:    Let $(\widehat{\mathbb{B}}_{i,h}\widehat{V}_{i,h+1}^\pi)(\cdot,\cdot), \Gamma_{i,h}(\cdot,\cdot) \leftarrow \mathbb{O}(\mathcal{D}_{i,h}, \widehat{V}_{i,h+1}^\pi, \delta)$
4:    Set $\widehat{Q}_{i,h}^\pi(\cdot,\cdot) \leftarrow \min\{H-h+1, (\widehat{\mathbb{B}}_{i,h}\widehat{V}_{i,h+1}^\pi)(\cdot,\cdot) - \Gamma_{i,h}(\cdot,\cdot)\}^+$
5:    Set $\widehat{V}_{i,h}^\pi(\cdot) \leftarrow \langle \widehat{Q}_{i,h}^\pi(\cdot,\cdot), \pi_h(\cdot|\cdot)\rangle_\mathcal{A}$
6: **end for**
7: **Return** $\widehat{V}_{i,1}^\pi(\cdot), \ldots, \widehat{V}_{i,H}^\pi(\cdot), \widehat{Q}_{i,1}^\pi(\cdot,\cdot), \ldots, \widehat{Q}_{i,H}^\pi(\cdot,\cdot)$.

---

**Algorithm 2** $\underline{P}$essimistic $\underline{E}$mpirical $\underline{R}$isk $\underline{M}$inimization (PERM)

---

**Require:** Offline dataset $\mathcal{D} = \{\mathcal{D}_i\}_{i=1}^n, \mathcal{D}_i := \{(x_{i,h}^\tau, a_{i,h}^\tau, r_{i,h}^\tau)_{h=1}^H\}_{\tau=1}^K$, policy class $\Pi$, confidence probability $\delta \in (0,1)$, a pessimistic offline policy evaluation algorithm **Evaluation** as a subroutine.

1: Set $\mathcal{D}_{i,h} = \{(x_{i,h}^\tau, a_{i,h}^\tau, r_{i,h}^\tau, x_{i,h+1}^\tau)\}_{\tau=1}^K$
2: $\pi^{\text{PERM}} = \arg\max_{\pi \in \Pi} \frac{1}{n}\sum_{i=1}^n \widehat{V}_{i,1}^\pi(x_1)$,
   where $[\widehat{V}_{i,1}^\pi(\cdot), \cdot, \ldots, \cdot] = $ **Evaluation**$\left(\{\mathcal{D}_{i,h}\}_{h=1}^H, \pi, \delta/(3nH\mathcal{N}_{(Hn)^{-1}}^\Pi)\right)$
3: **Return** $\pi^{\text{PERM}}$.

---

proposed offline RL with ZSG. To begin with, we consider a general individual MDP and an oracle $\mathbb{O}$, which returns us an empirical Bellman operator and an uncertainty quantifier, defined as follows.

**Definition 5 (Jin et al. 2021)** *For any individual MDP $M$, a dataset $\mathcal{D} \subseteq \mathcal{S} \times \mathcal{A} \times \mathcal{S} \times [0,1]$ that is compliant with $M$, a test function $V_\mathcal{D} \subseteq [0,H]^\mathcal{S}$ and a confidence level $\xi$, we have an oracle $\mathbb{O}(\mathcal{D}, V_\mathcal{D}, \xi)$ that returns $(\widehat{\mathbb{B}}V_\mathcal{D}(\cdot,\cdot), \Gamma(\cdot,\cdot))$, a tuple of Empirical Bellman operator and uncertainty quantifier, satisfying*

$$\mathbb{P}_\mathcal{D}\Big(\big|(\widehat{\mathbb{B}}V_\mathcal{D})(x,a) - (\mathbb{B}_M V_\mathcal{D})(x,a)\big|$$
$$\leq \Gamma(x,a) \text{ for all } (x,a) \in \mathcal{S} \times \mathcal{A}\Big) \geq 1 - \xi.$$

**Remark 6** *Here we adapt a test function $V_\mathcal{D}$ that can depend on the dataset $\mathcal{D}$ itself. Therefore, $\Gamma$ is a function that depends on both the dataset and the test function class. We do not specify the test function class in this definition, and we will discuss its specific realization in Section 6.*

**Remark 7** *For general non-linear MDPs, one may employ the bootstrapping technique to estimate uncertainty, in line with the bootstrapped DQN approach developed by (Osband et al., 2016). We note that when the bootstrapping method is straightforward to implement, the assumption of having access to an uncertainty quantifier is reasonable.*

Based on the oracle $\mathbb{O}$, we propose our pessimistic policy evaluation (PPE) algorithm as Algorithm 1. In general, PPE takes a given policy $\pi$ as its input, and its goal is to evaluate the V value and Q value $\{(V_{i,h}^\pi, Q_{i,h}^\pi)\}_{h=1}^H$ of $\pi$ on MDP $\mathcal{M}_i$. Since the agent is not allowed to interact with $\mathcal{M}_i$, PPE evaluates the value based on the offline dataset $\{\mathcal{D}_{i,h}\}_{h=1}^H$. At each stage $h$, PPE utilizes the oracle $\mathbb{O}$ and obtains the empirical Bellman operator based on $\mathcal{D}_{i,h}$ as well as its uncertainty quantifier, with high probability. Then PPE applies the *pessimism principle* to build the estimation of the Q function based on the empirical Bellman operator and the uncertainty quantifier. Such a principle has been widely studied and used in offline policy optimization, such as pessimistic value iteration (PEVI) (Jin et al., 2021). To

compare with, we use the pessimism principle in the policy evaluation problem.

**Remark 8** *In our framework, pessimism can indeed facilitate generalization, rather than hinder it. Specifically, we employ pessimism to construct reliable Q functions for each environment individually. This approach supports broader generalization by maintaining multiple Q-networks separately. By doing so, we ensure that each Q function is robust within its specific environment, while the collective set of Q functions enables the system to generalize across different environments.*

**5.2. Model-based approach: pessimistic empirical risk minimization**

Given PPE, we propose algorithms that have the ZSG ability. We first propose a pessimistic empirical risk minimization (PERM) method which is model-based and conceptually simple. The algorithm details are in Algorithm 2. In detail, for each dataset $\mathcal{D}_i$ drawn from $i$-th environments, PERM builds a model using PPE to evaluate the policy $\pi$ under the environment $\mathcal{M}_i$. Then PERM outputs a policy $\pi^{\text{PERM}} \in \Pi$ that maximizes the average pessimistic value, i.e., $1/n \sum_{i=1}^n \widehat{V}_{i,1}^\pi(x_1)$. Our approach is inspired by the classical empirical risk minimization approach adopted in supervised learning, and the Optimistic Model-based ERM proposed in Ye et al. (2023) for online RL. Our setting is more challenging than the previous ones due to the RL setting and the offline setting, where the interaction between the agent and the environment is completely disallowed. Therefore, unlike Ye et al. (2023), which adopted an optimism-style estimation to the policy value, we adopt a pessimism-style estimation to fight the distribution shift issue in the offline setting.

Next we propose a theoretical analysis of PERM. Denote $\mathcal{N}_\epsilon^\Pi$ as the $\epsilon$-covering number of the policy space $\Pi$ w.r.t. distance $d(\pi^1, \pi^2) = \max_{s \in \mathcal{S}, h \in [H]} \|\pi_h^1(\cdot|s) - \pi_h^2(\cdot|s)\|_1$. Then we have the following theorem to provide an upper bound of the suboptimality gap of the output policy $\pi^{\text{PERM}}$.

**Theorem 9** *Set the Evaluation subroutine in Algorithm 2*

*as PPE (Algo.1). Let $\Gamma_{i,h}$ be the uncertainty quantifier returned by $\mathbb{O}$ through the PERM. Then w.p. at least $1 - \delta$, the output $\pi^{PERM}$ of Algorithm 2 satisfies*

$$\text{SubOpt}(\pi^{PERM}) \leq \underbrace{7\sqrt{\frac{2\log(6\mathcal{N}^{\Pi}_{(Hn)^{-1}}/\delta)}{n}}}_{I_1:\text{Supervised learning (SL) error}}$$

$$+ \underbrace{\frac{2}{n}\sum_{i=1}^{n}\sum_{h=1}^{H}\mathbb{E}_{i,\pi^*}\left[\Gamma_{i,h}(s_h, a_h)|s_1 = x_1\right],}_{I_2:\text{Reinforcement learning (RL) error}} \quad (1)$$

*where $\mathbb{E}_{i,\pi^*}$ is w.r.t. the trajectory induced by $\pi^*$ with the transition $\mathcal{P}_i$ in the underlying MDP $\mathcal{M}_i$.*

**Proof** See Appendix B.1. ∎

**Remark 10** *The covering number $\mathcal{N}^{\Pi}_{(Hn)^{-1}}$ depends on the policy class $\Pi$. Without any specific assumptions, the policy class $\Pi$ that consists of all the policies $\pi = \{\pi_h\}_{h=1}^{H}, \pi_h : \mathcal{S} \mapsto \Delta(\mathcal{A})$ and the log $\epsilon$-covering number $\log\mathcal{N}^{\Pi}_{\epsilon} = O(|\mathcal{A}||\mathcal{S}|H\log(1 + |\mathcal{A}|/\epsilon))$.*

**Remark 11** *The SL error can be easily improved to a distribution-dependent bound $\log\mathcal{N} \cdot Var/\sqrt{n}$, where $\mathcal{N}$ is the covering number term denoted in $I_1$, $Var = \max_{\pi}\mathbb{V}_{c\sim C}V^{\pi}_{c,1}(x_1)$ is the variance of the context distribution, by using a Bernstein-type concentration inequality in our proof. Therefore, for the singleton environment case where $|C| = 1$, our suboptimality gap reduces to the one of PEVI in Jin et al. (2021).*

**Remark 12** *In real-world settings, as the number of sampled contexts $n$ may be very large, it is unrealistic to manage $n$ models simultaneously in the implementation of PERM algorithm, thus we provide the suboptimality bound in line with Theorem 9 when the offline dataset is merged into $m$ contexts such that $m < n$. See Theorem 28 in Appendix C.*

Theorem 9 shows that the ZSG gap of PERM is bounded by two terms $I_1$ and $I_2$. $I_1$, which we call *supervised learning error*, depends on the number of environments $n$ in the offline dataset $\mathcal{D}$ and the covering number of the function (policy) class, which is similar to the generalization error in supervised learning. $I_2$, which we call it *reinforcement learning error*, is decided by the optimal policy $\pi^*$ that achieves the best zero-shot generalization performance and the uncertainty quantifier $\Gamma_{i,h}$. In general, $I_2$ is the "intrinsic uncertainty" denoted by Jin et al. (2021) over $n$ MDPs, which characterizes how well each dataset $\mathcal{D}_i$ covers the optimal policy $\pi^*$.

## 5.3. Model-free approach: pessimistic proximal policy optimization

**Algorithm 3** Pessimistic Proximal Policy Optimzation (PPPO)

---

**Require:** Offline dataset $\mathcal{D} = \{\mathcal{D}_i\}_{i=1}^{n}, \mathcal{D}_i := \{(x^{\tau}_{i,h}, a^{\tau}_{i,h}, r^{\tau}_{i,h})_{h=1}^{H}\}_{\tau=1}^{K}$, confidence probability $\delta \in (0, 1)$, a pessimistic offline policy evaluation algorithm **Evaluation** as a subroutine.
1: Set $\mathcal{D}_{i,h} = \{(x^{\tau \cdot H+h}_{i,h}, a^{\tau \cdot H+h}_{i,h}, r^{\tau \cdot H+h}_{i,h}, x^{\tau \cdot H+h}_{i,h+1})\}_{\tau=0}^{\lfloor K/H \rfloor - 1}$
2: Set $\pi_{0,h}(\cdot|\cdot)$ as uniform distribution over $\mathcal{A}$ and $\widehat{Q}^{\pi_0}_{0,h}(\cdot, \cdot)$ as zero functions.
3: **for** $i = 1, 2, \cdots, n$ **do**
4:     Set $\pi_{i,h}(\cdot|\cdot) \propto \pi_{i-1,h}(\cdot|\cdot) \cdot \exp(\alpha \cdot \widehat{Q}^{\pi_{i-1}}_{i-1,h}(\cdot, \cdot))$
5:     Set $[\cdot, \ldots, \cdot, \widehat{Q}^{\pi_i}_{i,1}(\cdot, \cdot), \ldots, \widehat{Q}^{\pi_i}_{i,H}(\cdot, \cdot)] =$ **Evaluation**$(\{\mathcal{D}_{i,h}\}_{h=1}^{H}, \pi_i, \delta/(nH))$
6: **end for**
7: **Return** $\pi^{PPPO} = \text{random}(\pi_1, ..., \pi_n)$

---

PERM in Algorithm 2 works as a general model-based algorithm framework to enable ZSG for any pessimistic policy evaluation oracle. However, note that in order to implement PERM, one needs to maintain $n$ different models or critic functions simultaneously in order to evaluate $\sum_{i=1}^{n}\widehat{V}^{\pi}_{i,1}(x_1)$ for any candidate policy $\pi$. Note that existing online RL (Ghosh et al., 2021) achieves ZSG by a model-free approach, which only maintains $n$ policies rather than models/critic functions. Therefore, one natural question is whether we can design a *model-free* offline RL algorithm also with access only to policies.

We propose the pessimistic proximal policy optimization (PPPO) in Algorithm 3 to address this issue. Our algorithm is inspired by the optimistic PPO (Cai et al., 2020) originally proposed for online RL. PPPO also adapts PPE as its subroutine to evaluate any given policy pessimistically. Unlike PERM, PPPO only maintains $n$ policies $\pi_1, ..., \pi_n$, each of them is associated with an MDP $\mathcal{M}_n$ from the offline dataset. In detail, PPPO assigns an order for MDPs in the offline dataset and names them $\mathcal{M}_1, ..., \mathcal{M}_n$. For $i$-th MDP $\mathcal{M}_i$, PPPO selects the $i$-th policy $\pi_i$ as the solution of the proximal policy optimization starting from $\pi_{i-1}$, which is

$$\pi_i \leftarrow \underset{\pi}{\text{argmax}}\, V^{\pi}_{i-1,1}(x_1)$$
$$- \alpha^{-1}\mathbb{E}_{i-1,\pi_{i-1}}[\text{KL}(\pi\|\pi_{i-1})|s_1 = x_1], \quad (2)$$

where $\alpha$ is the step size parameter. Since $V^{\pi}_{i-1,1}(x_1)$ is not achievable, we use a linear approximation $L_{i-1}(\pi)$ to replace $V^{\pi}_{i-1,1}(x_1)$, where

$$L_{i-1}(\pi) = V^{\pi_{i-1}}_{i-1,1}(x_1) + \mathbb{E}_{i-1,\pi_{i-1}}\Bigg[$$
$$\sum_{h=1}^{H}\langle\widehat{Q}^{\pi_{i-1}}_{i-1,h}(x_h, \cdot), \pi_h(\cdot|x_h) - \pi_{i-1,h}(\cdot|x_h)\rangle\bigg|s_1 = x_1\Bigg], \quad (3)$$

where $\widehat{Q}^{\pi_{i-1}}_{i-1,h} \approx Q^{\pi_{i-1}}_{i-1,h}$ are the Q values evaluated on the offline dataset for $\mathcal{M}_{i-1}$. (2) and (3) give us a close-form

solution of $\pi$ in Line 4 in Algorithm 3. Such a routine corresponds to one iteration of PPO (Schulman et al., 2017). Finally, PPPO outputs $\pi^{\text{PPPO}}$ as a random selection from $\pi_1, ..., \pi_n$.

**Remark 13** *In Algorithm 3, we adopt a data-splitting trick (Jin et al., 2021) to build $\mathcal{D}_{i,h}$, where we only utilize each trajectory once for one data tuple at some stage h. It is only used to avoid the statistical dependency of $\widehat{V}_{i,h+1}^{\pi_i}(\cdot)$ and $x_{i,h+1}^{\tau}$ for the purpose of theoretical analysis.*

The following theorem bounds the suboptimality of PPPO.

**Theorem 14** *Set the Evaluation subroutine in Algorithm 3 as Algorithm 1. Let $\Gamma_{i,h}$ be the uncertainty quantifier returned by $\mathbb{O}$ through the PPPO. Selecting $\alpha = 1/\sqrt{H^2 n}$. Then selecting $\delta = 1/8$, w.p. at least $2/3$, we have*

$$SubOpt(\pi^{\text{PPPO}}) \le 10 \Bigg( \underbrace{\sqrt{\frac{\log |\mathcal{A}| H^2}{n}}}_{I_1 : SL \text{ error}}$$

$$+ \underbrace{\frac{1}{n}\sum_{i=1}^{n}\sum_{h=1}^{H} \mathbb{E}_{i,\pi^*}\left[\Gamma_{i,h}(s_h, a_h)|s_1 = x_1\right]}_{I_2 : RL \text{ error}} \Bigg).$$

*where $\mathbb{E}_{i,\pi^*}$ is w.r.t. the trajectory induced by $\pi^*$ with the transition $\mathcal{P}_i$ in the underlying MDP $\mathcal{M}_i$.*

**Proof** See Appendix B.2. ∎

**Remark 15** *As in Remark 12, we also provide the suboptimality bound in line with Theorem 14 when the offline dataset is merged into m contexts such that $m < n$. See Theorem 29 in Appendix C.*

Theorem 14 shows that the suboptimality gap of PPPO can also be bounded by the SL error $I_1$ and RL error $I_2$. Interestingly, $I_1$ in Theorem 14 for PPPO only depends on the cardinality of the action space $|\mathcal{A}|$, which is different from the covering number term in $I_1$ for PERM. Such a difference is due to the fact that PPPO outputs the final policy $\pi^{\text{PPPO}}$ as a random selection from $n$ existing policies, while PERM outputs one policy $\pi^{\text{PERM}}$. Whether these two guarantees can be unified into one remains an open question.

## 6. Provable generalization for offline linear MDPs

In this section, we instantiate our Algo.2 and Algo.3 for general MDPs on specific MDP classes. We consider the linear MDPs defined as follows.

**Assumption 16 (Yang and Wang 2019; Jin et al. 2019)** *We assume $\forall i \in C, \mathcal{M}_i$ is a linear MDP with a known feature map $\phi : \mathcal{S} \times \mathcal{A} \to \mathbb{R}^d$ if there exist d unknown*

measures $\mu_{i,h} = (\mu_{i,h}^{(1)}, \ldots, \mu_{i,h}^{(d)})$ over $\mathcal{S}$ and an unknown vector $\theta_{i,h} \in \mathbb{R}^d$ such that

$$P_{i,h}(x' \mid x, a) = \langle \phi(x,a), \mu_{i,h}(x') \rangle,$$
$$\mathbb{E}\left[r_{i,h}(s_h, a_h) \mid s_h = x, a_h = a\right] = \langle \phi(x,a), \theta_{i,h} \rangle \quad (4)$$

*for all $(x, a, x') \in \mathcal{S} \times \mathcal{A} \times \mathcal{S}$ at every step $h \in [H]$. We assume $\|\phi(x,a)\| \le 1$ for all $(x, a) \in \mathcal{S} \times \mathcal{A}$ and $\max\{\|\mu_{i,h}(\mathcal{S})\|, \|\theta_{i,h}\|\} \le \sqrt{d}$ at each step $h \in [H]$, and we define $\|\mu_{i,h}(\mathcal{S})\| = \int_{\mathcal{S}} \|\mu_{i,h}(x)\| \, \mathrm{d}x$.*

We first specialize the general PPE algorithm (Algo.1) to obtain the PPE algorithm tailored for linear MDPs (Algo.4). This specialization is achieved by constructing $\widehat{\mathbb{B}}_{i,h}\widehat{V}_{i,h+1}^{\pi}$, $\Gamma_{i,h}$, and $\widehat{V}_{i,h}^{\pi}$ based on the dataset $\mathcal{D}_i$. We denote the set of trajectory indexes in $\mathcal{D}_{i,h}$ as $\mathcal{B}_{i,h}$. Algo.4 subsequently functions as the policy evaluation subroutine in Algo.2 and Algo.3 for linear MDPs. In detail, we construct $\widehat{\mathbb{B}}_{i,h}\widehat{V}_{i,h+1}$ (which is the estimation of $\mathbb{B}_{i,h}\widehat{V}_{i,h+1}$) as $(\widehat{\mathbb{B}}_{i,h}\widehat{V}_{i,h+1})(x,a) = \phi(x,a)^\top \widehat{w}_{i,h}$, where

$$\widehat{w}_{i,h} = \operatorname{argmin}_{w \in \mathbb{R}^d} \sum_{\tau \in \mathcal{B}_{i,h}} \left(r_{i,h}^\tau + \widehat{V}_{i,h+1}(x_{i,h}^{-,\tau})\right.$$
$$\left. - \phi(x_{i,h}^\tau, a_{i,h}^\tau)^\top w\right)^2 + \lambda \cdot \|w\|_2^2 \quad (5)$$

with $\lambda > 0$ being the regularization parameter. The closed-form solution to (5) is in Line 4 in Algorithm 4. Besides, we construct the uncertainty quantifier $\Gamma_{i,h}$ based on $\mathcal{D}_i$ as

$$\Gamma_{i,h}(x, a) = \beta(\delta) \cdot \|\phi(x,a)\|_{\Lambda_{i,h}^{-1}}, \Lambda_{i,h}$$
$$= \sum_{\tau \in \mathcal{B}_{i,h}} \phi(x_{i,h}^\tau, a_{i,h}^\tau)\phi(x_{i,h}^\tau, a_{i,h}^\tau)^\top + \lambda \cdot I,$$

with $\beta(\delta) > 0$ being the scaling parameter.

---

**Algorithm 4** $\underline{P}$essimistic $\underline{P}$olicy $\underline{E}$valuation (PPE): Linear MDP

**Require:** Offline dataset $\{\mathcal{D}_{i,h}\}_{h=1}^H, \mathcal{D}_{i,h} = \{(x_{i,h}^\tau, a_{i,h}^\tau, r_{i,h}^\tau, x_{i,h}^{-,\tau})\}_{\tau \in \mathcal{B}_{i,h}}$, policy $\pi$, confidence probability $\delta \in (0, 1)$.
1: Initialize $\widehat{V}_{i,H+1}^\pi(\cdot) \leftarrow 0, \forall i \in [n]$.
2: **for** step $h = H, H-1, \ldots, 1$ **do**
3:      Set $\Lambda_{i,h} \leftarrow \sum_{\tau \in \mathcal{B}_{i,h}} \phi(x_{i,h}^\tau, a_{i,h}^\tau)\phi(x_{i,h}^\tau, a_{i,h}^\tau)^\top + \lambda \cdot I$.
4:      Set $\widehat{w}_{i,h} \leftarrow \Lambda_{i,h}^{-1}(\sum_{\tau \in \mathcal{B}_{i,h}} \phi(x_{i,h}^\tau, a_{i,h}^\tau) \cdot (r_{i,h}^\tau + \widehat{V}_{i,h+1}^\pi(x_{i,h}^{-,\tau})))$.
5:      Set $\Gamma_{i,h}(\cdot, \cdot) \leftarrow \beta(\delta) \cdot (\phi(\cdot, \cdot)^\top \Lambda_{i,h}^{-1}\phi(\cdot, \cdot))^{1/2}$.
6:      Set $\widehat{Q}_{i,h}^\pi(\cdot, \cdot) \leftarrow \min\{\phi(\cdot, \cdot)^\top \widehat{w}_{i,h} - \Gamma_{i,h}(\cdot, \cdot), H - h + 1\}^+$.
7:      Set $\widehat{V}_{i,h}^\pi(\cdot) \leftarrow \langle \widehat{Q}_{i,h}^\pi(\cdot, \cdot), \pi_h(\cdot|\cdot)\rangle_{\mathcal{A}}$
8: **end for**
9: **Return** $\widehat{V}_{i,1}^\pi(\cdot), \ldots, \widehat{V}_{i,H}^\pi(\cdot), \widehat{Q}_{i,1}^\pi(\cdot, \cdot), \ldots, \widehat{Q}_{i,H}^\pi(\cdot, \cdot)$.

---

The following theorem shows the suboptimality gaps for Algo.2 (utilizing subroutine Algo.4) and Algo.3 (also with subroutine Algo.4).

**Theorem 17** *Under Assumption 16, in Algorithm 4, we set* $\lambda = 1, \quad \beta(\delta) = c \cdot dH\sqrt{\log(2dHK/\delta)}$, *where* $c > 0$ *is a positive constant. Then, we have:*
*(i) for the output policy* $\pi^{PERM}$ *of Algo.2 with subroutine Algo.4, w.p. at least* $1 - \delta$, *the suboptimality gap satisfies*

$$SubOpt(\pi^{PERM}) \leq 7\sqrt{\frac{7\log(6\mathcal{N}^{\Pi}_{(Hn)^{-1}}/\delta)}{n} + \frac{2\beta(\frac{\delta}{3nH\mathcal{N}^{\Pi}_{(Hn)^{-1}}})}{n}}$$

$$\cdot \sum_{i=1}^{n}\sum_{h=1}^{H}\mathbb{E}_{i,\pi^*}\left[\|\phi(s_h, a_h)\|_{\widetilde{\Lambda}^{-1}_{i,h}} \,\big|\, s_1 = x_1\right], \tag{6}$$

*(ii) for the output policy* $\pi^{PPPO}$ *of Algo.3 with subroutine Algo.4, setting* $\delta = 1/8$, *then with probability at least* $2/3$, *the suboptimality gap satisfies*

$$\text{SubOpt}(\pi^{PPPO}) \leq 10\left(\sqrt{\frac{\log|\mathcal{A}|H^2}{n}} + \frac{\beta(\frac{1}{4nH})}{n}\right.$$

$$\left. \cdot \sum_{i=1}^{n}\sum_{h=1}^{H}\mathbb{E}_{i,\pi^*}\left[\|\phi(s_h, a_h)\|_{\bar{\Lambda}^{-1}_{i,h}} \,\big|\, s_1 = x_1\right]\right), \tag{7}$$

*where* $\mathbb{E}_{i,\pi^*}$ *is with respect to the trajectory induced by* $\pi^*$ *with the transition* $\mathcal{P}_i$ *in the underlying MDP* $\mathcal{M}_i$ *given the fixed matrix* $\widetilde{\Lambda}_{i,h}$ *or* $\bar{\Lambda}_{i,h}$.

$\|\phi(s_h, a_h)\|_{\Lambda^{-1}_{i,h}}$ indicates how well the state-action pair $(s_h, a_h)$ is covered by the dataset $\mathcal{D}_i$. The term $\sum_{i=1}^{n}\sum_{h=1}^{H}\mathbb{E}_{i,\pi^*}\left[\|\phi(s_h, a_h)\|_{\Lambda^{-1}_{i,h}} \,\big|\, s_1 = x_1\right]$ in the suboptimality gap in Theorem 17 is small if for each context $i \in [n]$, the dataset $\mathcal{D}_i$ well covers the trajectory induced by the optimal policy $\pi^*$ on the corresponding MDP $\mathcal{M}_i$.

**Well-explored behavior policy** Next we consider a case where the dataset $\mathcal{D}$ consists of i.i.d. trajectories collecting from different environments. Suppose $\mathcal{D}$ consists of $n$ independent datasets $\mathcal{D}_1, \ldots, \mathcal{D}_n$, and for each environment $i$, $\mathcal{D}_i$ consists of $K$ trajectories $\mathcal{D}_i = \{(x^{\tau}_{i,h}, a^{\tau}_{i,h}, r^{\tau}_{i,h})^{H}_{h=1}\}^{K}_{\tau=1}$ independently and identically induced by a fixed behavior policy $\bar{\pi}_i$ in the linear MDP $\mathcal{M}_i$. We have the following assumption on well-explored policy:

**Definition 18 (Duan et al. 2020; Jin et al. 2021)** *For an behavior policy* $\bar{\pi}$ *and an episodic linear MDP* $\mathcal{M}$ *with feature map* $\phi$, *we say* $\bar{\pi}$ *well-explores* $\mathcal{M}$ *with constant* $c$ *if there exists an absolute positive constant* $c > 0$ *such that*

$$\forall h \in [H], \lambda_{\min}(\Sigma_h) \geq c/d,$$

$$\text{where } \Sigma_h = \mathbb{E}_{\bar{\pi},\mathcal{M}}\left[\phi(s_h, a_h)\phi(s_h, a_h)^{\top}\right].$$

A well-explored policy guarantees that the obtained trajectories is "uniform" enough to represent any policy and value function. The following corollary shows that with the above assumption, the suboptimality gaps of Algo.2 (with subroutine Algo.4) and Algo.3 (with subroutine Algo.4) decay to 0 when $n$ and $K$ are large enough.

**Corollary 19** *Suppose that for each* $i \in [n]$, $\mathcal{D}_i$ *is generated by behavior policy* $\bar{\pi}_i$ *which well-explores MDP* $\mathcal{M}_i$ *with constant* $c_i \geq c_{min}$. *In Algo.4, we set* $\lambda = 1, \beta(\delta) = c' \cdot dH\sqrt{\log(4dHK/\delta)}$ *where* $c' > 0$ *is a positive constant. Suppose we have* $K \geq 40d/c_{min}\log(4dnH/\delta)$ *and set* $C^*_n := 1/n \cdot \sum_{i=1}^{n}c_i^{-1/2}$. *Then we have:*
*(i) for the output* $\pi^{PERM}$ *of Algo.2 with subroutine Algo.4, w.p. at least* $1 - \delta$, *the suboptimality gap satisfies*

$$SubOpt(\pi^{PERM}) \leq 7\sqrt{\frac{2\log(6\mathcal{N}^{\Pi}_{(Hn)^{-1}}/\delta)}{n}}$$

$$+ 2\sqrt{2}c' \cdot d^{3/2}H^2K^{-1/2}\sqrt{\log(12dHnK\mathcal{N}^{\Pi}_{(Hn)^{-1}}/\delta)} \cdot C^*_n, \tag{8}$$

*(ii) for the output policy* $\pi^{PPPO}$ *of Algo.3 with subroutine Algo.4, setting* $\delta = 1/8$, *then with probability at least* $2/3$, *the suboptimality gap satisfies*

$$\text{SubOpt}(\pi^{PPPO}) \leq 10\left(\sqrt{\frac{\log|\mathcal{A}|H^2}{n}}\right.$$

$$\left. + 2\sqrt{2}c' \cdot d^{3/2}H^{2.5}K^{-1/2}\sqrt{\log(16dHnK)} \cdot C^*_n\right). \tag{9}$$

**Remark 20** *The mixed coverage parameter* $C^*_n = \frac{1}{n}\sum_{i=1}^{n}\frac{1}{\sqrt{c_i}}$ *is small if for any* $i \in [n]$, $c_i$ *is large, i.e., the minimum eigenvalue of* $\Sigma_{i,h} = \mathbb{E}_{\bar{\pi}_i,\mathcal{M}_i}\left[\phi(s_h, a_h)\phi(s_h, a_h)^{\top}\right]$ *is large. Note that* $\lambda_{min}(\Sigma_{i,h})$ *indicates how well the behavior policy* $\bar{\pi}_i$ *explores the state-action pairs on MDP* $\mathcal{M}_i$; *this shows that if for each environment* $i \in [n]$, *the behavior policy explores* $\mathcal{M}_i$ *well, the suboptimality gap will be small.*

**Remark 21** *Under the same conditions of Corollary 19:*
*(i) If* $n \geq \frac{392\log(6\mathcal{N}^{\Pi}_{(Hn)^{-1}}/\delta)}{\epsilon^2}$ *and* $K \geq \max\{\frac{40d}{c_{min}}\log(\frac{4dnH}{\delta}), \frac{32c'^2d^3H^4\log(12dHnK\mathcal{N}^{\Pi}_{(Hn)^{-1}}/\delta)C^{*2}_n}{\epsilon^2}\}$, *then w.p. at least* $1 - \delta$, $SubOpt(\pi^{PERM}) \leq \epsilon$.
*(ii) If* $n \geq \frac{400H^2\log(|\mathcal{A}|)}{\epsilon^2}$ *and* $K \geq \max\{\frac{40d}{c_{min}}\log(16dnH), \frac{32c'^2d^3H^5\log(16dHnK)C^{*2}_n}{\epsilon^2}\}$, *then w.p. at least* $2/3$, $SubOpt(\pi^{PPPO}) \leq \epsilon$.

Corollary 19 suggests that both of our proposed algorithms enjoy the $O(n^{-1/2} + K^{-1/2} \cdot C^*_n)$ convergence rate to the optimal policy $\pi^*$ given a well-exploration data collection assumption, where $C^*_n$ is a mixed coverage parameter over $n$ environments defined in Corollary 19.

## 7. Conclusion and Future Work

In this work, we study the zero-shot generalization (ZSG) performance of offline reinforcement learning (RL). We propose two offline RL frameworks, pessimistic empirical risk minimization and pessimistic proximal policy optimization, and show that both of them can find the optimal policy with ZSG ability. We also show that such a generalization

property does not hold for offline RL without knowing the context information of the environment, which demonstrates the necessity of our proposed new algorithms. Currently, our theorems and algorithm design depend on the i.i.d. assumption of the environment selection. How to relax such an assumption remains an interesting future direction.

## Impact Statement

This paper presents work whose goal is to advance the field of Machine Learning. There are many potential societal consequences of our work, none which we feel must be specifically highlighted here.

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

## A. Results in Section 4

### A.1. Proof of Proposition 4

Let $\mathcal{D}' = \{(x^\tau_{c_\tau,h}, a^\tau_{c_\tau,h}, r^\tau_{c_\tau,h})\}^{H,K}_{h=1,\tau=1}$ denote the merged dataset, where each trajectory belongs to a context $c_\tau$. For simplicity, let $\mathcal{D}_c$ denote the collection of trajectories that belong to MDP $\mathcal{M}_c$. Then each trajectory in $\mathcal{D}'$ is generated by the following steps:

- The experimenter randomly samples an environment $c \sim C$.

- The experimenter collect a trajectory from the episodic MDP $\mathcal{M}_c$.

Then for any $x', r', \tau$ we have

$$
\mathbb{P}_{\mathcal{D}'}(r^\tau_{c_\tau,h} = r', x^\tau_{c_\tau,h+1} = x' | \{(x^j_{c_j,h}, a^j_{c_j,h})\}^\tau_{j=1}, \{r^j_{c_j,h}, x^j_{c_j,h+1}\}^{\tau-1}_{j=1})
$$

$$
= \frac{\mathbb{P}_{\mathcal{D}'}(r^\tau_{c_\tau,h} = r', x^\tau_{c_\tau,h+1} = x', \{(x^j_{c_j,h}, a^j_{c_j,h})\}^\tau_{j=1}, \{r^j_{c_j,h}, x^j_{c_j,h+1}\}^{\tau-1}_{j=1})}{\mathbb{P}_{\mathcal{D}'}(\{(x^j_{c_j,h}, a^j_{c_j,h})\}^\tau_{j=1}, \{r^j_{c_j,h}, x^j_{c_j,h+1}\}^{\tau-1}_{j=1})}
$$

$$
= \sum_{c \in C} \mathbb{P}_{\mathcal{D}'}(r^\tau_{c_\tau,h} = r', x^\tau_{c_\tau,h+1} = x' | \{(x^j_{c_j,h}, a^j_{c_j,h})\}^\tau_{j=1}, \{r^j_{c_j,h}, x^j_{c_j,h+1}\}^{\tau-1}_{j=1}, c_\tau = c) q(c), \tag{10}
$$

where

$$
q(c') := \frac{\mathbb{P}_{\mathcal{D}'}(\{(x^j_{c_j,h}, a^j_{c_j,h})\}^\tau_{j=1}, \{r^j_{c_j,h}, x^j_{c_j,h+1}\}^{\tau-1}_{j=1}, c_\tau = c')}{\sum_{c \in C} \mathbb{P}_{\mathcal{D}'}(\{(x^j_{c_j,h}, a^j_{c_j,h})\}^\tau_{j=1}, \{r^j_{c_j,h}, x^j_{c_j,h+1}\}^{\tau-1}_{j=1}, c_\tau = c)}.
$$

Next, we further have

$$
\begin{aligned}
(10) \\
&= \sum_{c \in C} \mathbb{P}_c(r_{c,h}(s_h) = r', s_{h+1} = x' | s_h = x^\tau_{c_\tau,h}, a_h = a^\tau_{c_\tau,h}) q(c) \\
&= \sum_{c \in C} \frac{\mathbb{P}_c(r_{c,h}(s_h) = r', s_{h+1} = x' | s_h = x^\tau_{c_\tau,h}, a_h = a^\tau_{c_\tau,h}) \mathbb{P}_{\mathcal{D}'}(s_h = x^\tau_{c_\tau,h}, a_h = a^\tau_{c_\tau,h}, c_\tau = c)}{\sum_{c \in C} \mathbb{P}_{\mathcal{D}'}(s_h = x^\tau_{c_\tau,h}, a_h = a^\tau_{c_\tau,h}, c_\tau = c)} \\
&= \sum_{c \in C} p(c) \cdot \frac{\mathbb{P}_c(r_{c,h}(s_h) = r', s_{h+1} = x' | s_h = x^\tau_{c_\tau,h}, a_h = a^\tau_{c_\tau,h}) \mathbb{P}_c(s_h = x^\tau_{c_\tau,h}, a_h = a^\tau_{c_\tau,h})}{\sum_{c \in C} p(c) \cdot \mathbb{P}_c(s_h = x^\tau_{c_\tau,h}, a_h = a^\tau_{c_\tau,h})} \\
&= \mathbb{E}_{c \sim C} \frac{\mathbb{P}_c(r_{c,h}(s_h) = r', s_{h+1} = x' | s_h = x^\tau_{c_\tau,h}, a_h = a^\tau_{c_\tau,h}) \mu_{c,h}(x^\tau_{c_\tau,h}, a^\tau_{c_\tau,h})}{\mathbb{E}_{c \sim C} \mu_{c,h}(x^\tau_{c_\tau,h}, a^\tau_{c_\tau,h})},
\end{aligned}
$$

where the first equality holds since for all trajectories $\tau$ satisfying $c_\tau = c$, they are compliant with $\mathcal{M}_c$, the second one holds since all trajectories are independent of each other, the third and fourth ones hold due to the definition of $\mu_{c,h}(\cdot, \cdot)$.

## A.2. PEVI algorithm

---

**Algorithm 5** (Jin et al., 2021) Pessimistic Value Iteration (PEVI)

---

**Require:** Dataset $\mathcal{D} = \{(x_{c_\tau,h}^\tau, a_{c_\tau,h}^\tau, r_{c_\tau,h}^\tau)_{h=1}^H\}_{\tau=1}^K$, confidence probability $\delta \in (0,1)$.

1: Initialization: Set $\widehat{V}_{H+1}(\cdot) \leftarrow 0$.
2: **for** step $h = H, H-1, \ldots, 1$ **do**
3:     Set $\Lambda_h \leftarrow \sum_{\tau=1}^K \phi(x_h^\tau, a_h^\tau)\phi(x_h^\tau, a_h^\tau)^\top + \lambda \cdot I$.
4:     Set $\widehat{w}_h \leftarrow \Lambda_h^{-1}(\sum_{\tau=1}^K \phi(x_h^\tau, a_h^\tau) \cdot (r_h^\tau + \widehat{V}_{h+1}(x_{h+1}^\tau)))$.
5:     Set $\Gamma_h(\cdot, \cdot) \leftarrow \beta(\delta) \cdot (\phi(\cdot, \cdot)^\top \Lambda_h^{-1} \phi(\cdot, \cdot))^{1/2}$.
6:     Set $\widehat{Q}_h(\cdot, \cdot) \leftarrow \min\{\phi(\cdot, \cdot)^\top \widehat{w}_h - \Gamma_h(\cdot, \cdot), H - h + 1\}^+$.
7:     Set $\widehat{\pi}_h(\cdot \mid \cdot) \leftarrow \arg\max_{\pi_h} \langle \widehat{Q}_h(\cdot, \cdot), \pi_h(\cdot \mid \cdot)\rangle_{\mathcal{A}}$.
8:     Set $\widehat{V}_h(\cdot) \leftarrow \langle \widehat{Q}_h(\cdot, \cdot), \widehat{\pi}_h(\cdot \mid \cdot)\rangle_{\mathcal{A}}$.
9: **end for**
10: **Return** $\pi^{\text{PEVI}} = \{\widehat{\pi}_h\}_{h=1}^H$.

---

We analyze the suboptimality gap of the Pessimistic Value Iteration (PEVI) ((Jin et al., 2021)) in the contextual linear MDP setting without context information to demonstrate that by finding the optimal policy for $\bar{\mathcal{M}}$ is not enough to find the policy that performs well on MDPs with context information.

**Pessimistic Value Iteration (PEVI).** Let $\overline{\pi}^*$ be the optimal policy w.r.t. the average MDP $\bar{\mathcal{M}}$. We analyze the performance of the Pessimistic Value Iteration (PEVI) (Jin et al., 2021) under the unknown context information setting. The details of PEVI is in Algo.5.

Suppose that $\bar{\mathcal{D}}$ consists of $K$ number of trajectories generated i.i.d. following by a fixed behavior policy $\bar{\pi}$. Then the following theorem shows the suboptimality gap for Algo.5 does not converge to 0 even when the data size grows to infinity.

**Theorem 22** *Assume that $\bar{\pi}$ In Algo.4, we set*

$$\lambda = 1, \quad \beta(\delta) = c' \cdot dH\sqrt{\log(4dHK/\delta)}, \tag{11}$$

*where $c' > 0$ is a positive constant. Suppose we have $K \geq \widetilde{c} \cdot d\log(4dH/\xi)$, where $\widetilde{c} > 0$ is a sufficiently large positive constant that depends on $c$. Then we have: w.p. at least $1 - \delta$, for the output policy $\pi^{\text{PEVI}}$ of Algo.5,*

$$\sup_\pi V_{\bar{\mathcal{M}},1}^\pi - V_{\bar{\mathcal{M}},1}^{\pi^{\text{PEVI}}} \leq c'' \cdot d^{3/2}H^2K^{-1/2}\sqrt{\log(4dHK/\delta)}, \tag{12}$$

*and the suboptimality gap satisfies*

$$\text{SubOpt}(\pi^{\text{PEVI}}) \leq c'' \cdot d^{3/2}H^2K^{-1/2}\sqrt{\log(4dHK/\delta)} + 2\sup_\pi |V_{\bar{\mathcal{M}},1}^\pi(x_1) - \mathbb{E}_{c\sim C}V_{c,1}^\pi(x_1)|, \tag{13}$$

*where $c'' > 0$ is a positive constant that only depends on $c$ and $c'$.*

**Proof** [Proof of Theorem 22] First, we define the value function on the average MDP $\bar{\mathcal{M}}$ as follows.

$$\overline{V}_h^\pi(x) = \mathbb{E}_{\pi,\bar{\mathcal{M}}}\Big[\sum_{i=h}^H r_i(s_i, a_i) \,\big|\, s_h = x\Big]. \tag{14}$$

We then decompose the suboptimality gap as follows.

$$\begin{aligned}
&\text{SubOpt}(\pi^{\text{PEVI}}) \\
&= \mathbb{E}_{c\sim C}\big[V_{c,1}^{\pi^*}(x_1)\big] - \mathbb{E}_{c\sim C}\big[V_{c,1}^{\pi^{\text{PEVI}}}(x_1)\big] \\
&= \overline{V}_1^{\overline{\pi}^*}(x_1) - \overline{V}_1^{\pi^{\text{PEVI}}}(x_1) + \big(\mathbb{E}_{c\sim C}\big[V_{c,1}^{\pi^*}(x_1)\big] - \overline{V}_1^{\overline{\pi}^*}(x_1)\big) + \big(\overline{V}_1^{\pi^{\text{PEVI}}}(x_1) - \mathbb{E}_{c\sim C}\big[V_{c,1}^{\pi^{\text{PEVI}}}(x_1)\big]\big) \\
&\leq \overline{V}_1^{\overline{\pi}^*}(x_1) - \overline{V}_1^{\pi^{\text{PEVI}}}(x_1) + 2\sup_\pi |V_{\bar{\mathcal{M}},1}^\pi(x_1) - \mathbb{E}_{c\sim C}V_{c,1}^\pi(x_1)|. 
\end{aligned} \tag{15}$$

Then, applying Corollary 4.6 in (Jin et al., 2021), we can get that w.p. at least $1 - \delta$

$$\overline{V}_1^{\overline{\pi}^*}(x_1) - \overline{V}_1^{\pi^{\text{PEVI}}}(x_1) \leq c'' \cdot d^{3/2} H^2 K^{-1/2} \sqrt{\log(4dHK/\delta)}, \tag{16}$$

which, together with Eq.(15) completes the proof.

■

Theorem 22 shows that by adapting the standard pessimistic offline RL algorithm over the offline dataset without context information, the learned policy $\pi^{\text{PEVI}}$ converges to the optimal policy $\overline{\pi}^*$ over the average MDP $\overline{\mathcal{M}}$.

# B. Proof of Theorems in Section 5

## B.1. Proof of Theorem 9

We define the model estimation error as

$$\iota_{i,h}^\pi(x, a) = (\mathbb{B}_{i,h} \widehat{V}_{i,h+1}^\pi)(x, a) - \widehat{Q}_{i,h}^\pi(x, a). \tag{17}$$

And we define the following condition

$$\left| (\widehat{\mathbb{B}}_{i,h} \widehat{V}_{i,h+1}^\pi)(x, a) - (\mathbb{B}_{i,h} \widehat{V}_{i,h+1}^\pi)(x, a) \right| \leq \Gamma_{i,h}(x, a) \text{ for all } i \in [n], \pi \in \Pi, (x, a) \in \mathcal{S} \times \mathcal{A}, h \in [H]. \tag{18}$$

We introduce the following lemma to bound the model estimation error.

**Lemma 23 (Model estimation error bound (Adapted from Lemma 5.1 in (Jin et al., 2021))** *Under the condition of Eq.(18), we have*

$$0 \leq \iota_{i,h}^\pi(x, a) \leq 2\Gamma_{i,h}(x, a), \quad \text{for all } i \in [n], \pi \in \Pi, (x, a) \in \mathcal{S} \times \mathcal{A}, h \in [H]. \tag{19}$$

Then, we prove the following lemma for pessimism in V values.

**Lemma 24 (Pessimism for Estimated V Values)** *Under the condition of Eq.(18), for any $i \in [n], \pi \in \Pi, x \in \mathcal{S}$, we have*

$$V_{i,h}^\pi(x) \geq \widehat{V}_{i,h}^\pi(x). \tag{20}$$

**Proof** For any $i \in [n], \pi \in \Pi, x \in \mathcal{S}, a \in \mathcal{A}$, we have

$$\begin{aligned}
Q_{i,h}^\pi(x, a) &- \widehat{Q}_{i,h}^\pi(x, a) \\
&\geq r_{i,h}(x, a) + (\mathbb{B}_{i,h} V_{i,h+1}^\pi)(x, a) - \left( r_{i,h}(s, a) + (\widehat{\mathbb{B}}_{i,h} \widehat{V}_{i,h+1}^\pi)(x, a) - \Gamma_{i,h}(x, a) \right) \\
&= (\mathbb{B}_{i,h} V_{i,h+1}^\pi)(x, a) - (\mathbb{B}_{i,h} \widehat{V}_{i,h+1}^\pi)(x, a) + \Gamma_{i,h}(x, a) \\
&\quad - \left( (\widehat{\mathbb{B}}_{i,h} \widehat{V}_{i,h+1}^\pi)(x, a) - \mathbb{B}_{i,h} \widehat{V}_{i,h+1}^\pi)(x, a) \right) \\
&\geq (\mathbb{B}_{i,h} V_{i,h+1}^\pi)(x, a) - (\mathbb{B}_{i,h} \widehat{V}_{i,h+1}^\pi)(x, a) \\
&= \left( P_{i,h}(V_{i,h+1}^\pi - \widehat{V}_{i,h+1}^\pi) \right)(x, a),
\end{aligned}$$

where the second inequality is because of Eq.(18). And since in the $H + 1$ step we have $V_{i,H+1}^\pi = \widehat{V}_{i,h+1}^\pi = 0$, we can get $Q_{i,H}^\pi(x, a) - \widehat{Q}_{i,H}^\pi(x, a)$. Then we use induction to prove $Q_{i,h}^\pi(x, a) \geq \widehat{Q}_{i,h}^\pi(x, a)$ for all $h$. Given $Q_{i,h+1}^\pi(x, a) \geq \widehat{Q}_{i,h+1}^\pi(x, a)$, we have

$$\begin{aligned}
Q_{i,h}^\pi(x, a) - \widehat{Q}_{i,h}^\pi(x, a) &\geq \left( P_{i,h}(V_{i,h+1}^\pi - \widehat{V}_{i,h+1}^\pi) \right)(x, a) \\
&= \mathbb{E} \left[ \langle Q_{i,h+1}^\pi(s_{h+1}, \cdot) - \widehat{Q}_{i,h+1}^\pi(s_{h+1}, \cdot), \pi_{h+1}(\cdot|s_{h+1}) \rangle_\mathcal{A} | s_h = x, a_h = a \right] \\
&\geq 0. \tag{21}
\end{aligned}$$

Then we have

$$V_{i,h}^{\pi}(x) - \widehat{V}_{i,h}^{\pi}(x) = \langle Q_{i,h}^{\pi}(x,\cdot) - \widehat{Q}_{i,h}^{\pi}(x,\cdot), \pi_h(\cdot\,|\,x)\rangle_{\mathcal{A}} \geq 0\,.$$

∎

Then we start our proof.

**Proof** [Proof of Theorem 9]

First, we decompose the suboptimality gap as follows

$$\begin{aligned}
&\text{SubOpt}(\pi^{\text{PERM}}) \\
&= \mathbb{E}_{c\sim C} V_{c,1}^{\pi^*}(x_1) - V_{c,1}^{\widehat{\pi}^*}(x_1) \\
&= \mathbb{E}_{c\sim C} V_{c,1}^{\pi^*}(x_1) - \frac{1}{n}\sum_{i=1}^{n} V_{i,1}^{\pi^*}(x_1) + \frac{1}{n}\sum_{i=1}^{n} V_{i,1}^{\pi^{\text{PERM}}}(x_1) - \mathbb{E}_{c\sim C} V_{c,1}^{\pi^{\text{PERM}}}(x_1) \\
&\quad + \frac{1}{n}\sum_{i=1}^{n}\left(V_{i,1}^{\pi^*}(x_1) - V_{i,1}^{\pi^{\text{PERM}}}(x_1)\right).
\end{aligned} \tag{22}$$

For the first two terms, we can bound them following the standard generalization techniques ((Ye et al., 2023)), *i.e.*, we use the covering argument, Chernoff bound, and union bound.

Define the distance between policies $d(\pi^1, \pi^2) \triangleq \max_{s\in\mathcal{S}, h\in[H]} \|\pi_h^1(\cdot|s) - \pi_h^2(\cdot|s)\|_1$. We construct the $\epsilon$-covering set $\widetilde{\Pi}$ w.r.t. $d$ such that

$$\forall \pi \in \Pi, \exists \widetilde{\pi} \in \widetilde{\Pi}, s.t. \quad d(\pi, \widetilde{\pi}) \leq \epsilon. \tag{23}$$

Then we have

$$\forall i \in [n], \pi \in \Pi, \exists \widetilde{\pi} \in \widetilde{\Pi}, s.t. V_{i,1}^{\pi}(x_1) - V_{i,1}^{\widetilde{\pi}}(x_1) \leq H\epsilon. \tag{24}$$

By the definition of the covering number, $\left|\widetilde{\Pi}\right| = \mathcal{N}_\epsilon^{\Pi}$. By Chernoff bound and union bound over the policy set $\widetilde{\Pi}$, we have with prob. at least $1 - \frac{\delta}{3}$, for any $\widetilde{\pi} \in \widetilde{\Pi}$,

$$\left|\frac{1}{n}\sum_{i=1}^{n} V_{i,1}^{\widetilde{\pi}}(x_1) - \mathbb{E}_{c\sim C} V_{c,1}^{\widetilde{\pi}}(x_1)\right| \leq \sqrt{\frac{2\log(6\mathcal{N}_\epsilon^{\Pi}/\delta)}{n}}. \tag{25}$$

By Eq.(24) and Eq.(25), $\forall i \in [n], \pi \in \Pi, \exists \widetilde{\pi} \in \widetilde{\Pi}$ with $\left|\widetilde{\Pi}\right| = \mathcal{N}_\epsilon^{\Pi}$, $s.t. V_{i,1}^{\pi}(x_1) - V_{i,1}^{\widetilde{\pi}}(x_1) \leq H\epsilon$, and with probability at least $1 - \delta/3$, we have

$$\begin{aligned}
&\left|\frac{1}{n}\sum_{i=1}^{n} V_{i,1}^{\pi}(x_1) - \mathbb{E}_{c\sim C} V_{c,1}^{\pi}(x_1)\right| \\
&\leq \left|\frac{1}{n}\sum_{i=1}^{n} V_{i,1}^{\widetilde{\pi}}(s_1) - \mathbb{E}_{c\sim C} V_{c,1}^{\widetilde{\pi}}(x_1)\right| \\
&\quad + \left|\frac{1}{n}\sum_{i=1}^{n} V_{i,1}^{\pi}(s_1) - \frac{1}{n}\sum_{i=1}^{n} V_{i,1}^{\widetilde{\pi}}(s_1)\right| + \left|\mathbb{E}_{c\sim C} V_{c,1}^{\widetilde{\pi}}(x_1) - \mathbb{E}_{c\sim C} V_{c,1}^{\pi}(x_1)\right| \\
&\leq \sqrt{\frac{2\log(6\mathcal{N}_\epsilon^{\Pi}/\delta)}{n}} + 2H\epsilon\,.
\end{aligned} \tag{26}$$

Therefore, we have for the first two terms, w.p. at least $1 - \frac{2}{3}\delta$ we can upper bound them with $4H\epsilon + 2\sqrt{\frac{2\log(6\mathcal{N}_\epsilon^{\Pi}/\delta)}{n}}$.

Then, what remains is to bound the term $\frac{1}{n}\sum_{i=1}^{n}\left(V_{i,1}^{\pi^*}(x_1) - V_{i,1}^{\pi^{\text{PERM}}}(x_1)\right)$.

First, by similar arguments, we have

$$
\begin{aligned}
V_{i,1}^{\pi^*}(x_1) - V_{i,1}^{\pi^{\text{PERM}}}(x_1) &\le \left(V_{i,1}^{\pi^*}(x_1) - V_{i,1}^{\widetilde{\pi}^{\text{PERM}}}(x_1)\right) + |V_{i,1}^{\widetilde{\pi}^{\text{PERM}}}(x_1) - V_{i,1}^{\pi^{\text{PERM}}}(x_1)| \\
&\le H\epsilon + V_{i,1}^{\pi^*}(x_1) - V_{i,1}^{\widetilde{\pi}^{\text{PERM}}}(x_1),
\end{aligned}
\tag{27}
$$

where $\widetilde{\pi}^{\text{PERM}} \in \widetilde{\Pi}$ such that $|V_{i,1}^{\widetilde{\pi}^{\text{PERM}}}(x_1) - V_{i,1}^{\pi^{\text{PERM}}}(x_1)| \le H\epsilon$.

By the definition of the oracle in Definition.5, the algorithm design of Algo.1 (e.g., we call oracle $\mathbb{O}(\mathcal{D}_h, \widehat{V}_{h+1}, \delta/(3nH\mathcal{N}_{(Hn)^{-1}}^{\Pi}))$, and use a union bound over $H$ steps, $n$ contexts, and $\mathcal{N}_{(Hn)^{-1}}^{\Pi}$ policies, we have: with probability at least $1 - \delta/3$, the condition in Eq.(18) holds (with the policy class $\Pi$ replaced by $\widetilde{\Pi}$ (and $\epsilon = 1/(Hn)$).

Then, we have

$$
\begin{aligned}
\frac{1}{n}\sum_{i=1}^{n}&\left(V_{i,1}^{\pi^*}(x_1) - V_{i,1}^{\widetilde{\pi}^{\text{PERM}}}(x_1)\right) \\
&\le \frac{1}{n}\sum_{i=1}^{n}\left(V_{i,1}^{\pi^*}(x_1) - \widehat{V}_{i,1}^{\widetilde{\pi}^{\text{PERM}}}(x_1)\right) \\
&= \frac{1}{n}\sum_{i=1}^{n}\left(V_{i,1}^{\pi^*}(x_1) - \widehat{V}_{i,1}^{\pi^{\text{PERM}}}(x_1)\right) + \frac{1}{n}\sum_{i=1}^{n}\left(\widehat{V}_{i,1}^{\pi^{\text{PERM}}}(x_1) - \widehat{V}_{i,1}^{\widetilde{\pi}^{\text{PERM}}}(x_1)\right) \\
&\le \frac{1}{n}\sum_{i=1}^{n}\left(V_{i,1}^{\pi^*}(x_1) - \widehat{V}_{i,1}^{\pi^{\text{PERM}}}(x_1)\right) + H \cdot \frac{1}{Hn} \\
&\le \frac{1}{n}\sum_{i=1}^{n}\left(V_{i,1}^{\pi^*}(x_1) - \widehat{V}_{i,1}^{\pi^*}(x_1)\right) + 1/n,
\end{aligned}
\tag{28}
$$

where the first inequality holds because of the pessimism in Lemma 24, the second inequality holds because $|\widehat{V}_{i,1}^{\widetilde{\pi}^{\text{PERM}}}(x_1) - \widehat{V}_{i,1}^{\pi^{\text{PERM}}}(x_1)| \le H\epsilon$ with $\epsilon$ here specified as $1/(Hn)$, and the last inequality holds because that in the algorithm design of Algo.2 we set $\pi^{\text{PERM}} = \text{argmax}_{\pi \in \Pi} \frac{1}{n}\sum_{i=1}^{n}\widehat{V}_{i,1}^{\pi}(x_1)$.

Then what left is to bound $V_{i,1}^{\pi^*}(x_1) - \widehat{V}_{i,1}^{\pi^*}(x_1)$.

And using Lemma A.1 in (Jin et al., 2021), we have

$$
\begin{aligned}
V_{i,1}^{\pi^*}(x_1) - \widehat{V}_{i,1}^{\pi^*}(x_1) &= -\sum_{h=1}^{H}\mathbb{E}_{\widehat{\pi}^*, \mathcal{M}_i}\left[\iota_{i,h}^{\pi^*}(s_h, a_h)\,\middle|\,s_1 = x\right] + \sum_{h=1}^{H}\mathbb{E}_{\pi^*, \mathcal{M}_i}\left[\iota_{i,h}^{\pi^*}(s_h, a_h)\,\middle|\,s_1 = x\right] \\
&\quad + \sum_{h=1}^{H}\mathbb{E}_{\pi^*, \mathcal{M}_i}\left[\langle\widehat{Q}_{i,h}^{\pi^*}(s_h, \cdot), \pi_h^*(\cdot\,|\,s_h) - \pi_h^*(\cdot\,|\,s_h)\rangle_{\mathcal{A}}\,\middle|\,s_1 = x\right] \\
&\le 2\sum_{h=1}^{H}\mathbb{E}_{\pi^*, \mathcal{M}_i}\left[\Gamma_{i,h}(s_h, a_h)\,\middle|\,s_1 = x\right],
\end{aligned}
\tag{29}
$$

where in the last inequality we use Lemma 23.

Finally, with Eq.(22), Eq.(26), Eq.(27), Eq.(28), and Eq.(29), with $\epsilon$ set as $\frac{1}{nH}$, we can get w.p. at least $1 - \delta$

$$\mathbb{E}_{c \sim C} V_{c,1}^{\pi^*}(x_1) - V_{c,1}^{\pi^{\mathrm{PERM}}}(x_1)$$

$$\leq \frac{5}{n} + 2\sqrt{\frac{2\log(6\mathcal{N}_{(Hn)^{-1}}^{\Pi}/\delta)}{n}} + \frac{2}{n} \sum_{i=1}^{n} \sum_{h=1}^{H} \mathbb{E}_{\pi^*, \mathcal{M}_i} \left[ \Gamma_{i,h}(s_h, a_h) | s_1 = x_1 \right]$$

$$\leq 7\sqrt{\frac{2\log(6\mathcal{N}_{(Hn)^{-1}}^{\Pi}/\delta)}{n}} + \frac{2}{n} \sum_{i=1}^{n} \sum_{h=1}^{H} \mathbb{E}_{\pi^*, \mathcal{M}_i} \left[ \Gamma_{i,h}(s_h, a_h) | s_1 = x_1 \right].$$

∎

### B.2. Proof of Theorem 14

Our proof has two steps. First, we define that

$$\iota_{i,h}(x,a) := \mathbb{B}_{i,h} V_{i,h+1}(x,a) - Q_{i,h}(x,a) \tag{30}$$

Then we have the following lemma from Jin et al. (2021):

**Lemma 25** *Define the event $\mathcal{E}$ as*

$$\mathcal{E} = \left\{ \left| (\widehat{\mathbb{B}} \widehat{V}_{i,h+1}^{\pi_i})(x,a) - (\mathbb{B}_{i,h} \widehat{V}_{i,h+1}^{\pi_i})(x,a) \right| \leq \Gamma_{i,h}(x,a) \ \forall (x,a) \in \mathcal{S} \times \mathcal{A}, \forall h \in [H], \forall i \in [n] \right\},$$

*Then by selecting the input parameter $\xi = \delta/(Hn)$ in $\mathbb{O}$, we have $\mathbb{P}(\mathcal{E}) \geq 1 - \delta$ and*

$$0 \leq \iota_{i,h}(x,a) \leq 2\Gamma_{i,h}(x,a).$$

**Proof** The proof is the same as [Lemma 5.1, Jin et al. 2021] with the probability assigned as $\delta/(Hn)$ and a union bound over $h \in [H], i \in [n]$. ∎

Next lemma shows the difference between the value of the optimal policy $\pi^*$ and number $n$ of different policies $\pi_i$ for $n$ MDPs.

**Lemma 26** *Let $\pi$ be an arbitrary policy. Then we have*

$$\sum_{i=1}^{n} [V_{i,1}^{\pi}(x_1) - V_{i,1}^{\pi^i}(x_1)] = \sum_{i=1}^{n} \sum_{h=1}^{H} \mathbb{E}_{i,\pi} [\langle Q_{i,h}(\cdot, \cdot), \pi_h(\cdot|\cdot) - \pi_{i,h}(\cdot|\cdot) \rangle_{\mathcal{A}}]$$

$$+ \sum_{i=1}^{n} \sum_{h=1}^{H} (\mathbb{E}_{i,\pi}[\iota_{i,h}(x_h, a_h)] - \mathbb{E}_{i,\pi_i}[\iota_{i,h}(x_h, a_h)]) \tag{31}$$

**Proof** The proof is the same as Lemma 3.1 in (Jin et al., 2021) except substituting $\pi$ into the lemma. ∎

We also have the following one-step lemma:

**Lemma 27 (Lemma 3.3, Cai et al. 2020)** *For any distribution $p^*, p \in \Delta(\mathcal{A})$, if $p'(\cdot) \propto p(\cdot) \cdot \exp(\alpha \cdot Q(x, \cdot))$, then*

$$\langle Q(x, \cdot), p^*(\cdot) - p(\cdot) \rangle \leq \alpha H^2/2 + \alpha^{-1} \cdot \left( KL(p^*(\cdot) \| p(\cdot)) - KL(p^*(\cdot) \| p'(\cdot)) \right).$$

Given the above lemmas, we begin our proof of Theorem 14.

**Proof** [Proof of Theorem 14] Combining Lemma 25 and Lemma 26, we have

$$\sum_{i=1}^{n}[V_{i,1}^{\pi^*}(x_1) - V_{i,1}^{\pi^i}(x_1)]$$

$$\leq \sum_{i=1}^{n}\sum_{h=1}^{H}\mathbb{E}_{i,\pi^*}[\langle Q_{i,h}, \pi_h^* - \pi_{i,h}\rangle] + 2\sum_{i=1}^{n}\sum_{h=1}^{H}\mathbb{E}_{i,\pi^*}[\Gamma_{i,h}(x_h, a_h)]$$

$$\leq \sum_{i=1}^{n}\sum_{h=1}^{H}\alpha H^2/2 + \alpha^{-1}\mathbb{E}_{i,\pi^*}[\text{KL}(\pi_h^*(\cdot|x_h)\|\pi_{i,h}(\cdot|x_h)) - \text{KL}(\pi_h^*(\cdot|x_h)\|\pi_{i+1,h}(\cdot|x_h))]$$

$$+ 2\sum_{i=1}^{n}\sum_{h=1}^{H}\mathbb{E}_{i,\pi^*}[\Gamma_{i,h}(x_h, a_h)]$$

$$\leq \alpha H^3 n/2 + \alpha^{-1}\cdot\sum_{h=1}^{H}\mathbb{E}_{i,\pi^*}[\text{KL}(\pi_h^*(\cdot|x_h)\|\pi_{1,h}(\cdot|x_h))] + 2\sum_{i=1}^{n}\sum_{h=1}^{H}\mathbb{E}_{i,\pi^*}[\Gamma_{i,h}(x_h, a_h)]$$

$$\leq \alpha H^3 n/2 + \alpha^{-1}H\log|A| + 2\sum_{i=1}^{n}\sum_{h=1}^{H}\mathbb{E}_{i,\pi^*}[\Gamma_{i,h}(x_h, a_h)],$$

where the last inequality holds since $\pi_{1,h}$ is the uniform distribution over $\mathcal{A}$. Then, selecting $\alpha = 1/\sqrt{H^2 n}$, we have

$$\sum_{i=1}^{n}[V_{i,1}^{\pi^*}(x_1) - V_{i,1}^{\pi^i}(x_1)] \leq 2\sqrt{n\log|A|H^2} + 2\sum_{i=1}^{n}\sum_{h=1}^{H}\mathbb{E}_{i,\pi^*}[\Gamma_{i,h}(s_h, a_h)],$$

which holds for the random selection of $\mathcal{D}$ with probability at least $1 - \delta$. Meanwhile, note that each MDP $M_i$ is drawn i.i.d. from $C$. Meanwhile, note that $\pi_i$ only depends on MDP $M_1, ..., M_{i-1}$. Therefore, by the standard online-to-batch conversion, we have

$$\mathbb{P}\left(\frac{1}{n}\sum_{i=1}^{n}[V_{i,1}^{\pi^*}(x_1) - V_{i,1}^{\pi_i}(x_1)] + \left(\frac{1}{n}\sum_{i=1}^{n}\mathbb{E}_{c\sim C}V_{c,1}^{\pi_i}(x_1) - \mathbb{E}_{c\sim C}V_{c,1}^{\pi^*}(x_1)\right) \leq 2H\sqrt{\frac{2\log 1/\delta}{n}}\right) \geq 1 - \delta,$$

which suggests that with probability at least $1 - 2\delta$,

$$\mathbb{E}_{c\sim C}V_{c,1}^{\pi^*}(x_1) - \frac{1}{n}\sum_{i=1}^{n}\mathbb{E}_{c\sim C}V_{c,1}^{\pi_i}(x_1) \leq 2\sqrt{\frac{\log|A|H^2}{n}} + \frac{2}{n}\sum_{i=1}^{n}\sum_{h=1}^{H}\mathbb{E}_{\pi^*}[\Gamma_{i,h}(x_h, a_h)] + 2\sqrt{\frac{2H\log 1/\delta}{n}}.$$

Therefore, by selecting $\pi^{\text{PPPO}} := \text{random}(\pi_1, ..., \pi_n)$ and applying the Markov inequality, setting $\delta = 1/8$, we have our bound holds. ∎

## C. Suboptimality bounds for real-world setups

In this section we state and prove the suboptimality bounds we promised in Remarks 12 and 15, where we merge the sampled contexts into $m$ groups (generally, $m << n$) to reduce the computational complexity in practical settings. The bound in Theorem 28 serves as a partial justification for the effectiveness of IQL-$m$V in our real-data experiments (Section **??**).

Assume $m|n$ and the $n$ contexts from offline dataset are equally partitioned into $m$ groups. We write the resulting average MDPs (see Proposition 4) for each group as $\bar{\mathcal{M}}_1, \ldots, \bar{\mathcal{M}}_m$. For each $\bar{\mathcal{M}}_j$, we regard it as an individual context in the sense of (18) and denote the resulting uncertainty quantifier and value function as $\Gamma'_{j,h}, V'^{\pi}_{j,h}$.

**Theorem 28 (Suboptimality bound for Remark 12)** *Assume the same setting as Theorem 9 with the original $n$ contexts grouped as $m$ contexts, and denote the resulting algorithm as PERM-$m$V. Then w.p. at least $1 - \delta$, the output $\pi'$ of PERM-$m$V satisfies*

$$\text{SubOpt}(\pi') \leq \underbrace{2\sqrt{\frac{2\log(6\mathcal{N}_{(Hm)^{-1}}^{\Pi}/\delta)}{n}}}_{I_1 : \text{Supervised learning (SL) error}} + \underbrace{\frac{2}{m}\sum_{j=1}^{m}\sum_{h=1}^{H}\mathbb{E}_{\pi^*,\bar{\mathcal{M}}_j}\left[\Gamma'_{j,h}(s_h, a_h)|s_1 = x_1\right]}_{I_2 : \text{Reinforcement learning (RL) error}}$$

$$\underbrace{+\frac{5}{m} + 2\sup_{\pi}\left|\frac{1}{n}\sum_{i=1}^{n}V_{i,1}^{\pi}(x_1) - \frac{1}{m}\sum_{j=1}^{m}V'^{\pi}_{j,1}(x_1)\right|}_{\text{Additional approximation error}},$$

where $\mathbb{E}_{j,\pi^*}$ is w.r.t. the trajectory induced by $\pi^*$ with the transition $\bar{\mathcal{P}}_j$ in the underlying average MDP $\bar{\mathcal{M}}_j$.

**Proof** [Proof of Theorem 28]

Similar to the proof of Theorem 9, we decompose the suboptimality gap as follows

$$\text{SubOpt}(\pi')$$
$$= \mathbb{E}_{c\sim C}V_{c,1}^{\pi^*}(x_1) - V_{c,1}^{\pi'}(x_1)$$
$$= \mathbb{E}_{c\sim C}V_{c,1}^{\pi^*}(x_1) - \frac{1}{n}\sum_{i=1}^{n}V_{i,1}^{\pi^*}(x_1) + \frac{1}{n}\sum_{i=1}^{n}V_{i,1}^{\pi'}(x_1) - \mathbb{E}_{c\sim C}V_{c,1}^{\pi'}(x_1)$$
$$+ \frac{1}{n}\sum_{i=1}^{n}V_{i,1}^{\pi^*}(x_1) - \frac{1}{m}\sum_{j=1}^{m}V'^{\pi^*}_{j,1}(x_1) + \frac{1}{m}\sum_{j=1}^{m}V'^{\pi'}_{j,1}(x_1) - \frac{1}{n}\sum_{i=1}^{n}V_{i,1}^{\pi'}(x_1)$$
$$+ \frac{1}{m}\sum_{j=1}^{m}\left(V'^{\pi^*}_{j,1}(x_1) - V'^{\pi'}_{j,1}(x_1)\right). \tag{32}$$

Note that we can bound the first and third lines of (32) with the exactly same arguments as the proof of Theorem 9, the only notation-wise difference is that the uncertainty quantifier becomes $\Gamma'$ as we are operating on the level of average MDP $\bar{\mathcal{M}}_j$.

The only thing left is to bound the second line of (32). This is the same in spirit of the bound (15), so that we can express the bound as follows

$$\frac{1}{n}\sum_{i=1}^{n}V_{i,1}^{\pi^*}(x_1) - \frac{1}{m}\sum_{j=1}^{m}V'^{\pi^*}_{j,1}(x_1) + \frac{1}{m}\sum_{j=1}^{m}V'^{\pi'}_{j,1}(x_1) - \frac{1}{n}\sum_{i=1}^{n}V_{i,1}^{\pi'}(x_1)$$
$$\leq 2\sup_{\pi}\left|\frac{1}{n}\sum_{i=1}^{n}V_{i,1}^{\pi}(x_1) - \frac{1}{m}\sum_{j=1}^{m}V'^{\pi}_{j,1}(x_1)\right|.$$

To conclude, our final bound can be expressed as: with $\epsilon$ set as $\frac{1}{mH}$, we can get w.p. at least $1 - \delta$

$$\text{SubOpt}(\pi')$$
$$\leq 2\sqrt{\frac{2\log(6\mathcal{N}_{(Hm)^{-1}}^{\Pi}/\delta)}{n}} + \frac{2}{m}\sum_{j=1}^{m}\sum_{h=1}^{H}\mathbb{E}_{\pi^*,\bar{\mathcal{M}}_j}\left[\Gamma'_{j,h}(s_h, a_h)|s_1 = x_1\right]$$
$$+ \frac{5}{m} + 2\sup_{\pi}\left|\frac{1}{n}\sum_{i=1}^{n}V_{i,1}^{\pi}(x_1) - \frac{1}{m}\sum_{j=1}^{m}V'^{\pi}_{j,1}(x_1)\right|.$$

$\blacksquare$

To prove the suboptimality bound for Remark 15, we denote that the policies produced by PPPO after merging dataset to $m$ groups to be $\pi_1, \ldots, \pi_m$, and the original PPPO algorithm would produce the policies as $\pi'_1, \ldots, \pi'_n$. We assume that the merging of dataset from $n$ to $m$ groups is only to combine the consecutive $n/m$ terms from $\pi'_1, \ldots, \pi'_n$ and preserves the order.

**Theorem 29 (Suboptimality bound for Remark 15)** *Assume the same setting as Theorem 14 with the original $n$ contexts grouped as $m$ contexts, and denote the resulting algorithm as PPPO-mV. Let $\Gamma'_{j,h}$ be the uncertainty quantifier returned by $\mathbb{O}$ through the PPPO-$mV$ algorithm. Selecting $\alpha = 1/\sqrt{H^2 m}$. Then selecting $\delta = 1/8$, w.p. at least $2/3$, we have*

$$SubOpt(\pi^{PPPO-mV}) \leq 10\bigg( \underbrace{\sqrt{\frac{\log|\mathcal{A}|H^2}{m}}}_{I_1:SL\ error} + \underbrace{\frac{1}{m}\sum_{j=1}^{m}\sum_{h=1}^{H} \mathbb{E}_{j,\pi^*}\left[\Gamma'_{j,h}(s_h,a_h)|s_1=x_1\right]}_{I_2:RL\ error}$$

$$+ \sup_{\pi}\left|\frac{1}{n}\sum_{i=1}^{n} V_{i,1}^{\pi}(x_1) - \frac{1}{m}\sum_{j=1}^{m} V'^{\pi}_{j,1}(x_1)\right| + \frac{1}{n}\sum_{i=1}^{n}\sup_{\pi}\left|\mathbb{E}_c[V_{c,1}^{\pi}(x_1)] - V_{i,1}^{\pi}(x_1)\right|$$

$$+ \frac{1}{m}\sum_{j=1}^{m}\sup_{\pi}\left|\mathbb{E}_c[V'^{\pi}_{c,1}(x_1)] - V'^{\pi}_{j,1}(x_1)\right| \bigg).$$

*where $\mathbb{E}_{j,\pi^*}$ is w.r.t. the trajectory induced by $\pi^*$ with the transition $\bar{\mathcal{P}}_j$ in the underlying MDP $\bar{\mathcal{M}}_j$.*

**Proof** [Proof of Theorem 29]

Using the same arguments as in the proof of Theorem 14 with $\alpha = 1/\sqrt{H^2 m}$, we can derive the bound

$$\sum_{j=1}^{m}[V'^{\pi^*}_{j,1}(x_1) - V'^{\pi_j}_{j,1}(x_1)] \leq 2\sqrt{m\log|A|H^2} + 2\sum_{j=1}^{m}\sum_{h=1}^{H}\mathbb{E}_{j,\pi^*}[\Gamma'_{j,h}(s_h,a_h)].$$

Leveraging this bound and online-to-batch, we obtain the following estimation

$$\mathbb{E}_c[V_{c,1}^{\pi^*}(x_1)] - \frac{1}{m}\sum_{j=1}^{m}\mathbb{E}_c[V_{c,1}^{\pi_j}(x_1)]$$

$$=\mathbb{E}_c[V_{c,1}^{\pi^*}(x_1)] - \frac{1}{n}\sum_{i=1}^{n}\mathbb{E}_c[V_{c,1}^{\pi'_i}(x_1)] + \frac{1}{n}\sum_{i=1}^{n}\mathbb{E}_c[V_{c,1}^{\pi'_i}(x_1)] - \frac{1}{m}\sum_{j=1}^{m}\mathbb{E}_c[V_{c,1}^{\pi_j}(x_1)]$$

$$\leq 2H\sqrt{\frac{2\log 1/\delta}{n}} + \frac{1}{n}\sum_{i=1}^{n}\left(\mathbb{E}_c[V_{c,1}^{\pi'_i}(x_1)] - V_{i,1}^{\pi'_i}(x_1)\right) + \frac{1}{n}\sum_{i=1}^{n} V_{i,1}^{\pi^*}(x_1) - \frac{1}{m}\sum_{j=1}^{m}\mathbb{E}_c[V_{c,1}^{\pi_j}(x_1)]$$

$$=2H\sqrt{\frac{2\log 1/\delta}{n}} + \frac{1}{n}\sum_{i=1}^{n} V_{i,1}^{\pi^*}(x_1) - \frac{1}{m}\sum_{j=1}^{m} V'^{\pi^*}_{j,1}(x_1)$$

$$+ \frac{1}{m}\sum_{j=1}^{m} V'^{\pi^*}_{j,1}(x_1) - \frac{1}{m}\sum_{j=1}^{m} V'^{\pi_j}_{j,1}(x_1)$$

$$+ \frac{1}{n}\sum_{i=1}^{n}\left(\mathbb{E}_c[V_{c,1}^{\pi'_i}(x_1)] - V_{i,1}^{\pi'_i}(x_1)\right) + \frac{1}{m}\sum_{j=1}^{m} V'^{\pi_j}_{j,1}(x_1) - \frac{1}{m}\sum_{j=1}^{m}\mathbb{E}_c[V_{c,1}^{\pi_j}(x_1)]$$

$$\leq 2H\sqrt{\frac{2\log 1/\delta}{n}} + \sup_{\pi}\left|\frac{1}{n}\sum_{i=1}^{n} V_{i,1}^{\pi}(x_1) - \frac{1}{m}\sum_{j=1}^{m} V'^{\pi}_{j,1}(x_1)\right|$$

$$+ 2\sqrt{\frac{\log|A|H^2}{m}} + \frac{2}{m}\sum_{j=1}^{m}\sum_{h=1}^{H}\mathbb{E}_{j,\pi^*}[\Gamma'_{j,h}(s_h,a_h)]$$

$$+ \frac{1}{n}\sum_{i=1}^{n}\sup_{\pi}\left|\mathbb{E}_c[V_{c,1}^{\pi}(x_1)] - V_{i,1}^{\pi}(x_1)\right| + \frac{1}{m}\sum_{j=1}^{m}\sup_{\pi}\left|\mathbb{E}_c[V'^{\pi}_{c,1}(x_1)] - V'^{\pi}_{j,1}(x_1)\right|.$$

Finally we apply Markov inequality and take $\delta = 1/8$ as in the proof of Theorem 14. ∎

# D. Results in Section 6

### D.1. Proof of Theorem 17

By (Jin et al., 2021), the parameters specified as $\lambda = 1$, $\quad \beta(\delta) = c \cdot dH\sqrt{\log(2dHK/\delta)}$, and applying union bound, we can get: for Algo.4, with probability at least $1 - \delta/3$

$$\left|(\widehat{\mathbb{B}}_{i,h}\widehat{V}_{i,h+1}^{\pi})(x,a) - (\mathbb{B}_{i,h}\widehat{V}_{i,h+1}^{\pi})(x,a)\right| \le \beta\big(\frac{\delta}{3nH\mathcal{N}_{(Hn)^{-1}}^{\Pi}}\big)\big(\phi(x,a)^{\top}\Lambda_{i,h}^{-1}\phi(x,a)\big)^{1/2},$$

$$\text{for all } i \in [n], \pi \in \widetilde{\Pi}, (x,a) \in \mathcal{S} \times \mathcal{A}, h \in [H], \tag{33}$$

where $\widetilde{\Pi}$ is the $\frac{1}{Hn}$-covering set of the policy space $\Pi$ w.r.t. distance $\mathrm{d}(\pi^1, \pi^2) = \max_{s \in \mathcal{S}, h \in [H]} \|\pi_h^1(\cdot|s) - \pi_h^2(\cdot|s)\|_1$.

Therefore, we can specify the $\Gamma_{i,h}(\cdot,\cdot)$ in Theorem 9 with $\beta\big(\frac{\delta}{3nH\mathcal{N}_{(Hn)^{-1}}^{\Pi}}\big)\big(\phi(x,a)^{\top}\Lambda_{i,h}^{-1}\phi(x,a)\big)^{1/2}$, and follow the same process as the proof of Theorem 9 to get the result for Algo.2 with subroutine Algo.4.

Similarly, we can get: we can get: for Algo.4, with probability at least $1 - 1/4$

$$\left|(\widehat{\mathbb{B}}_{i,h}\widehat{V}_{i,h+1})(x,a) - (\mathbb{B}_{i,h}\widehat{V}_{i,h+1})(x,a)\right| \le \beta\big(\frac{\delta}{4nH}\big)\big(\phi(x,a)^{\top}\Lambda_{i,h}^{-1}\phi(x,a)\big)^{1/2},$$

$$\text{for all } i \in [n], (x,a) \in \mathcal{S} \times \mathcal{A}, h \in [H]. \tag{34}$$

Therefore, we can specify the $\Gamma_{i,h}(\cdot,\cdot)$ in Theorem 14 with $\beta\big(\frac{\delta}{4nH}\big)\big(\phi(x,a)^{\top}\Lambda_{i,h}^{-1}\phi(x,a)\big)^{1/2}$ and follow the same process as the proof of Theorem 14 to get the result for Algo.3 with subroutine Algo.4.

### D.2. Proof of Corollary 19

By the assumption that $\mathcal{D}_i$ is generated by behavior policy $\bar{\pi}_i$ which well-explores MDP $\mathcal{M}_i$ with constant $c_i$ (where the well-explore is defined in Def.18), the proof of Corollary 4.6 in (Jin et al., 2021), and applying a union bound over $n$ contexts, we have that for Algo.2 with subroutine Algo.4 w.p. at least $1 - \delta/2$

$$\|\phi(x,a)\|_{\Lambda_{i,h}^{-1}} \le \sqrt{\frac{2d}{c_i K}}$$

$$\text{for all } i \in [n], (x,a) \in \mathcal{S} \times \mathcal{A} \text{ and all } h \in [H], \tag{35}$$

and for Algo.2 with subroutine Algo.4 w.p. at least $1 - \delta/2$

$$\|\phi(x,a)\|_{\Lambda_{i,h}^{-1}} \le \sqrt{\frac{2dH}{c_i K}}$$

$$\text{for all } i \in [n], (x,a) \in \mathcal{S} \times \mathcal{A} \text{ and all } h \in [H], \tag{36}$$

because we use the data splitting technique and we only utilize each trajectory once for one data tuple at some stage $h$, so we replace $K$ with $K/H$.

Then, the result follows by plugging the results above into Theorem17.

