# OpenReview forum: "Provable Zero-Shot Generalization in Offline Reinforcement Learning"
_ICML.cc/2025/Conference — ICML 2025 poster_

### Official Review · Reviewer_rcHz · 2025-03-12

**Overall Recommendation:** 2

**Summary:**

This paper investigates the generalization performance of policies learned in offline reinforcement learning (ORL) on test environments, specifically focusing on zero-shot performance in the average sense across test environments. The paper demonstrates that standard ORL methods, without access to contextual information, cannot possess the zero-shot generalization property (ZSG). To address this issue, the paper proposes a method based on contextual Markov Decision Processes (MDPs). This method leverages contextual information to partition the data into multiple MDPs and learns a policy that performs well on average across these MDPs. Theoretically, the paper proves that this method can learn a policy with ZSG.

**Claims And Evidence:**

1. In my opinion, the proposed method looks like a straightforward extension on the previous work [Is pessimism provably efficient for offline rl?], i.e., it extends the pessimism to the contextual MDPs. The authors should provide more discussion about the difference between these two works (e.g., more theoretical properties and the behaviors of the learned policies), and do not be confined to the comparison for the theoretical bounds.

2.  The context information should be given in the dataset, i.e., the dataset is divided by the contexts. This is hard to satisfy in pracetice. How will the algorithm behave in offline Contextual MDP if no context information is given?

**Essential References Not Discussed:**

n/a

**Experimental Designs Or Analyses:**

no experimtal results are provided.

**Methods And Evaluation Criteria:**

yes

**Other Comments Or Suggestions:**

n/a

**Other Strengths And Weaknesses:**

no experimental results provided, so that the feasibility and the effectiveness of the proposed algorithms are unkown.

**Questions For Authors:**

1. in line 157, the optimal policy is defined to be the one with highest expection value over the underlying MDPs, what about these MDPs are quite different? If they are different enough, an optimal policy may be different from MDP to MDP, which makes the average policy less useful. In other words, how to define the scope of generalization for zero shot learning?
2. the need of contextual information seems to be a major limitation, how to relax that requirement?

**Relation To Broader Scientific Literature:**

1. The considered situation is like the settings of domain generation where a few domains are given but no information about the new domain is given, and the task is to learn from the training domain a policy that is generalizable to the new domain. This paper addressed a similar problem under the offline RL settings and proposed two algorithms for it and provided theroretical analsis on that.

2. The relation with in-context learning RL or a few shot learning RL should also be discussed.

**Theoretical Claims:**

n/a

---

> ### Author Rebuttal · Authors · 2025-04-01
>
> Thanks very much for reviewing our work and for the valuable advice. Our responses are as follows.
>
> **Q1:**  Differences from (Jin et al. 2021).
>
> **A1:**  We respectfully argue that our work is **substantially different** from (Jin et al., 2021), both in **algorithmic design** and **theoretical analysis**. The core challenge we address is *how to generalize across different environments*, rather than simply applying pessimism within each contextual MDP.
> - **Algorithmic Design:** While both works leverage the pessimism principle, we use it only as a **building block**. Specifically, Algo.1 performs pessimistic evaluation per environment, but this is only a **subroutine** for Algo.2 and Algo.3, which focus on **generalization across environments**. To our knowledge, this is the **first method** to adapt pessimism from time-step-wise to **environment-wise updates** for generalization. Jin et al. (2021), by contrast, operates within a single MDP and does not address this challenge.
> - **Theoretical Analysis:** In contrast to Jin et al. (2021), which analyzes a single MDP, we study the ZSG problem over multiple environments. For example, the proof of Theorem 9 (PERM) requires carefully separating SL and RL errors across environments (Eq. 22–28). Our analyses are non-trivial and require substantial innovation to handle generalization across distinct datasets. Additionally, Section 4 shows that the naive application of standard pessimistic offline RL in Jin et al. (2021) can yield **drastically suboptimal behavior**. This highlights that **simple extensions of existing pessimistic RL algorithms fail** in the ZSG setting.
>
>
> **Q2:**  About the context information:
>
> **A2:**  We would like to clarify that our algorithms do not require access to the context itself, nor does it rely on any semantic information. Instead, it only needs a **context indicator**—a label or identifier used to group trajectories that originate from the same environment, which often will be saved in the original dataset. In such cases, identifying the origin of each trajectory (i.e., grouping by environment) is often feasible. For this reason, we respectfully argue that this assumption is practical in many real-world scenarios.
>
>
>
> That said, **relaxing the requirement** is an exciting future direction. One possibility is to infer latent environment clusters from trajectory data via representation learning which introduces challenges like cluster identifiability in the offline setting. We will clarify these points and expand the discussion in the final version, following your thoughtful suggestions.
>
>
>
> **Q3:**  No experiments.
>
> **A3:**  Thank you for the thoughtful suggestion. We would like to respectfully emphasize that this is a theoretical paper submitted to the **theory category**, and our primary goal is to develop first-principles theoretical insights into zero-shot generalization in offline RL—a significantly under-explored area compared to standard offline RL in a single environment or meta-RL approaches that rely on additional adaptation data. We believe that our theoretical contributions represent a substantial advancement in the foundations of reinforcement learning and are valuable in their own right. We plan to run a thorough experimental evaluation of our proposed PERM and PPPO algorithms in future work.
>
> **Q4:**  The relation to in-context or few-shot RL.
>
> **A4:**  In-context or few-shot RL typically assumes access to a small number of test-domain episodes or relies on history-dependent policies to enable **online adaptation**. In contrast, our setting focuses on zero-shot generalization, where **no additional interaction or adaptation** is allowed at test time. This necessitates learning a policy that performs well out of the box in any newly sampled environment from the context distribution.
>
> **Q5:**  If the MDPs are quite different, an optimal policy might differ from one MDP to another. How does one define zero-shot generalization meaningfully?
>
> **A5:**  In the ZSG setting, our goal is finding a single policy $\pi^*$ that maximizes expected return across the distribution of environments:
> $\pi^* = \arg \max_{\pi} E_c[V_{c,1}^{\pi}(x_1)]$. While $\pi^*$ may not be optimal in every individual environment, it achieves the best average performance, which is **theoretically the strongest achievable guarantee without test-time context**. This objective follows prior work (Ye et al., 2023), which proves that achieving near-optimality in each environment (i.e., $E_c[V_c^{\pi^*(c)}-V_c^{\pi}]\leq \epsilon$) is impractical in the ZSG setting. We also note that this setup is meaningful and well-aligned with how generalization is defined in supervised learning: data points (contexts) are drawn from a distribution, and the goal is to learn a single predictor (policy) that performs well on average over that distribution. Allowing additional interaction with the environment makes our problem shift to meta-RL, which is beyond our ZSG scope.

---

### Official Review · Reviewer_qPEC · 2025-03-13

**Overall Recommendation:** 4

**Summary:**

The work theoretically analyzes contextual reinforcement learning in the offline setting for zero-shot generalizability. In particular, the work proposes two algorithmic frameworks with provable zero-shot generalization abilities.


## update after rebuttal
I have read all reviews and am keeping my score. I do believe that the work is interesting to a good chunk of the (RL sub)community and would lead to interesting discussions on acceptance.

**Claims And Evidence:**

The claims in the paper are well supported by thorough and sound theoretical analysis.

**Essential References Not Discussed:**

To me, one essential literature that is missing is "Contextualize Me -- The case for context in RL" (https://openreview.net/forum?id=Y42xVBQusn). This work discusses the contextual RL case at length and proved that optimal policies require context. Figure 1 of the work under review also reminded me of a similar figure in the paper by Benjamins et al. Though, this work views the problem of ZSG in RL from the classical online setting. Besides the theoretical component, that work proposes multiple contextual environments and shows the optimality gap for various different agent types empirically.

**Experimental Designs Or Analyses:**

N/A

**Methods And Evaluation Criteria:**

N/A

**Other Comments Or Suggestions:**

Line 316: "... which we call **it** *reinforcement* ..." --> "... which we call *reinforcement* ..."

**Other Strengths And Weaknesses:**

The work is generally well written, though with the variety of sub/super-scripts it can get difficult to properly track every detail.
To the best of my knowledge the work is the first providing a theoretical analysis of ZSG in offline RL.

**Questions For Authors:**

* In line131 it is stated that the same starting state is assumed for each $\mathcal{M}_c$. Why is that needed?
* Do your proofs require assumptions about the "quality" of the context or are they independent of these?

**Relation To Broader Scientific Literature:**

I believe the work puts itself well into the broader scientific literature and not only focusing on ZSG RL literature

**Theoretical Claims:**

I have checked the correctness of the proofs in the appendices A.1 and B.1 while only skimming the proofs in B.2 and C.

One point I remain uncertain about is assumptions about the context at hand. To me the work it seems tho implicitly assume "perfect context information". Under this assumption, the theoretical analysis makes perfect sense to me, though it would probably be best to highlight this assumption.

---

> ### Author Rebuttal · Authors · 2025-04-01
>
> Thanks very much for your positive feedback and your valuable suggestions. Our responses are as follows.
>
> **Q1:**  One point I remain uncertain about is the assumptions about the context at hand. The work seems to implicitly assume perfect context information. It would be best to highlight this assumption.
>
> **A1:**  Thank you for raising this important point. We respectfully clarify that our approach does not assume access to perfect or rich context information. Specifically, our algorithm and analysis only require that, during training, we can identify which trajectories come from the same environment. That is, trajectories from the same environment must be grouped together under a shared context label—this label can be **arbitrary** and need not encode any meaningful or descriptive features of the environment. It merely serves to distinguish between different environment instances during training.
>
> Importantly, **no context information is required at test time**. The agent is expected to generalize to a new environment without observing any context label or side information.
>
> As shown in Section 4, without such grouping, the data effectively collapses into an average MDP, which leads to poor zero-shot generalization—a limitation of prior offline RL methods. We agree that this assumption could be more explicitly stated and will revise the paper to clarify it. Investigating whether this weak form of context grouping can be removed or relaxed further is an exciting direction for future work.
>
> **Q2:**  An essential reference missing is “Contextualize Me—The Case for Context in RL” by Benjamins et al., which discusses contextual RL extensively and also provides an online zero-shot analysis.
>
> **A2:**  Thank you for pointing out this important reference. We agree that the work by Benjamins et al. offers a valuable perspective on contextual RL and online zero-shot generalization. We will incorporate a discussion of this paper in the Related Work section to better situate our contributions and emphasize the broader significance of context in RL.
>
> **Q3:**  The notations are a bit heavy.
>
> **A3:**  Thank you for the helpful suggestion. We acknowledge that the variety of subscripts and superscripts—especially for multi-environment and time-step indexing—can become difficult to track. In the revision, we will include a notational summary table or add brief reminders in the text to improve readability. We also appreciate you pointing out minor typos, such as the phrase on line 316 (“... which we call it reinforcement ...”), and will correct them accordingly.
>
> **Q4:**  Same starting state is assumed. Is it necessary?
>
> **A4:**  Thank you for pointing this out. The assumption of a single starting state is made primarily for notational and explanatory simplicity, and it is without loss of generality. This assumption is common in the offline RL theory literature (e.g., [1]) and simplifies the presentation without affecting the core results. Our analysis can be extended to allow different initial states, as long as they are drawn from the same distribution. The main logic and proof techniques remain valid under this extension.
>
> To avoid unnecessary technical clutter, we chose to adopt the single-start assumption in line with prior theoretical works. We will clarify this in the paper by adding a footnote, as per your helpful suggestion.
>
> [1] Provably Efficient Reinforcement Learning with Rich Observations via Latent State Decoding, ICML 2019.
>
> **Q5:**  Do the theorems place restrictions on the quality of the context?
>
> **A5:**  Thank you for the thoughtful question. Our theorems do not place any restrictions on the quality or informativeness of the context, beyond requiring that each context label $i$ is drawn i.i.d. from the same underlying context distribution $C$. The context labels need only distinguish environments from one another—they do not need to encode any semantic or structured information.
>
> Even if some environments differ significantly in difficulty or dynamics, the final policy is optimized to perform as well as possible on average across environments sampled from $C$. We will expand the discussion in the paper to clarify this point, following your helpful suggestion.
>
> Again, thank you very much for your positive feedback and valuable suggestions. Please feel free to reach out with any further questions or comments.

---

### Official Review · Reviewer_KVto · 2025-03-14

**Overall Recommendation:** 2

**Summary:**

Main findings&Contribution:
1. Problem finding: Classic offline reinforcement learning methods often struggle to generalize unseen environments, primarily due to dataset coverage limitations and the absence of contextual information during training. Merging multi-environment datasets without preserving contextual distinctions may result in convergence to an "average MDP," resulting in poor zero-shot generalization (ZSG).
2. Main contribution: This paper proposes Pessimistic Empirical Risk Minimization (PERM) and Pessimistic Proximal Policy Optimization (PPPO). These methods leverage pessimistic evaluation(PPE) to balance data fitting and generalization.
3. Theoretical Guarantees: The suboptimality gap of PERM/PPPO is jointly determined by the supervised learning (SL) error (governed by the number of training environments n and dataset size) and the reinforcement learning (RL) error (influenced by offline data coverage quality and uncertainty quantifiers).

**Claims And Evidence:**

**1. Claim:** Standard offline RL fails at ZSG without context information because the dataset resembles an "average MDP."

**Evidence:** Proposition 4 theoretically shows that merging datasets across contexts creates an indistinguishable average MDP. A 2-context example (Figure 1) illustrates how optimal policies for individual environments differ from those for the average MDP.

​​**2. Claim:** PERM and PPPO achieve ZSG with suboptimality bounds decomposed into SL and RL errors.

**Evidence:** Theorems 9 and 14 bound the suboptimality gap for both algorithms. The bounds depend on policy class complexity (SL term) and dataset coverage (RL term). While the focus is theoretical, even simple experiments (e.g., tabular MDPs or linear function approximation) could validate the practical relevance of the bounds and the necessity of pessimism.

**3. ​Claim:** Pessimism enables generalization by preventing overfitting to individual environments.

​**Evidence:** The paper argues that per-environment pessimism avoids conflating dynamics across contexts. This is intuitively justified but not empirically validated. The theory assumes environments are i.i.d., which is critical for the SL error term but not thoroughly motivated (e.g., no discussion of non-i.i.d. settings).

**Essential References Not Discussed:**

The related work has been well-included in the main text of the paper.

**Experimental Designs Or Analyses:**

This paper does not provide experimental validation for the proposed methods.

**Methods And Evaluation Criteria:**

The methods are theoretically grounded and address the core challenges of ZSG in offline RL. The evaluation criteria are appropriate for a theory-focused paper but would benefit from broader empirical validation to demonstrate practical utility.

**Other Comments Or Suggestions:**

No other comments and suggestions.

**Other Strengths And Weaknesses:**

While the paper provides a rigorous theoretical analysis of ZSG in offline RL, the empirical validation of the proposed methods (PERM and PPPO) is limited in scope. To strengthen the practical relevance and credibility of the claims, the authors should augment their evaluation with ​cross-task generalization experiments on widely adopted benchmarks like D4RL[1].

[1] Fu, J., Kumar, A., Nachum, O., Tucker, G., & Levine, S. (2020). D4rl: Datasets for deep data-driven reinforcement learning. arXiv preprint arXiv:2004.07219.

**Questions For Authors:**

Q1. The reinforcement learning error term in the suboptimality gap scales linearly with the state-action space dimension $d$. For high-dimensional tasks, could this dimensional dependency result in significant performance degradation in practice? Have you considered incorporating representation learning (such as low-rank feature extraction) to mitigate the curse of dimensionality?

**Relation To Broader Scientific Literature:**

By integrating pessimism with multi-environment policy optimization, the paper bridges gaps between offline RL, generalization theory, and contextual decision-making, offering a principled framework for ZSG with implications for real-world applications where environment-specific interaction is costly or impossible.

**Theoretical Claims:**

The theoretical claims are ​largely correct and build thoughtfully on existing offline RL frameworks. The proofs leverage established techniques (pessimism, covering numbers, elliptical potentials) while extending them to the multi-environment ZSG setting. However, tighter characterization of environment diversity (beyond i.i.d. assumptions) and empirical validation would elevate the work from a theoretical contribution to a practical guide for ZSG in offline RL.

---

> ### Author Rebuttal · Authors · 2025-03-26
>
> Thanks very much for reviewing our work and for the valuable suggestions. Our responses are as follows.
>
> **Q1:**  Augment theory with experiments; necessity of pessimism
>
> **A1:** Thank you for the thoughtful suggestion. We would like to respectfully emphasize that this is a theoretical paper submitted to the theory category, and our primary goal is to develop first-principles theoretical insights into zero-shot generalization in offline RL—a significantly under-explored area compared to standard offline RL in a single environment or meta-RL approaches that rely on additional adaptation data. We believe that our theoretical contributions represent a substantial advancement in the foundations of reinforcement learning and are valuable in their own right. We plan to run a thorough experimental evaluation of our proposed PERM and PPPO algorithms in future work.
>
> For the necessity of pessimism, we conducted a small‐scale experiment on a Combination Lock environment with 10 contexts adapted from (Bose et. al, 2024). We use 500 exploratory trajectories from (Agarwal et. al, 2023) per context to form the offline dataset.
>
> The table reports the average returns of the learned policies (evaluated over 100 runs) for our PPPO algorithm (with and without pessimism). We see that standard PPPO achieves a higher average reward, validating our theoretical premise that using environment‐specific pessimism can improve zero‐shot generalization.
> | PPPO           | PPPO w/o Pessimism |
> |--------------------|------------------------|
> | 0.0650 ± 0.0173    | 0.0603 ± 0.008         |
>
>
> Alekh Agarwal, Yuda Song, Wen Sun, Kaiwen Wang, Mengdi Wang, and Xuezhou Zhang. Provable benefits of representational transfer in reinforcement learning. In The Thirty Sixth Annual Conference on Learning Theory, pages 2114–2187. PMLR, 2023.321
>
>
>
>
>
> **Q2:**  The paper argues that per-environment pessimism avoids conflating dynamics across contexts. This is intuitively justified but not empirically validated. The theory assumes environments are i.i.d., which is critical for the SL error term but not thoroughly motivated (e.g., no discussion of non-i.i.d. settings).
>
> **A2:**  Thank you for the insightful feedback. Our per-environment pessimism approach ensures that each environment’s offline data contributes to a separate (pessimistic) critic or model component. This preserves environment-specific information, which is leveraged for generalization—either by maximizing the averaged value (PERM) or conducting PPO-like updates across environments (PPPO). By focusing pessimism on individual environments, we avoid the pitfalls of simply averaging across distinct dynamics and rewards, which could undermine generalization, as we theoretically demonstrate in Section 4. Importantly, this approach is not only intuitive but also forms the basis for the theoretical guarantees presented in Theorems 9 and 14. Additionally, as mentioned in **A1**, we have conducted some empirical analysis to support these findings.
>
> Regarding the i.i.d. assumption, we acknowledge that generalization in offline RL under distribution shifts remains an open challenge, and we are happy to discuss more in revision. Prior work on RL generalization in the online setting (Ye et al., 2023) also assumes i.i.d. environments and highlights OOD generalization as an open question. We believe starting with the i.i.d. assumption—as done in prior work—is a reasonable first step, and we view extending this analysis to non-i.i.d. settings as an exciting future direction.
>
>
> **Q3:**  The RL error term in the suboptimality gap scales linearly with the state–action space dimension  $d$. Could this dimensional dependency result in significant performance degradation in practice? Have you considered incorporating representation learning (e.g., low-rank feature extraction) to mitigate the curse of dimensionality?
>
> **A3:**  We believe you are referring to Theorem 17, which analyzes the special case of linear MDPs. We respectfully clarify that the parameter $d$ here refers to the dimension of the linear feature map, as defined in Assumption 16, and **not** to the cardinality of the state space $|\mathcal{S}|$ or action space $|\mathcal{A}|$. In fact, this formulation inherently assumes a low-dimensional representation, where the true dynamics and rewards lie in a $d$-dimensional linear subspace. Without such an assumption, we agree with you that incorporating representation learning is necessary. Note that our work could incorporate with recent representation learning works (Ishfaq et al., 2024) and taking the low-dimensional learned representation as our features.
> We will clarify this point further in our revision to avoid any confusion following your advice.
>
> Again, thank you very much for reviewing our paper and for the thoughtful feedback. We hope our responses adequately address your concerns, and we would be happy to clarify further if needed.

---

### Decision · Program_Chairs · 2025-05-01

**Decision:**

Accept (poster)

**Comment:**

The reviews of this paper illustrates the growing desire for applicability of ideas to actual data. Two reviewers weren't interested in the paper just because of its theoretical nature.
Yet (qPEC) found it interesting to discuss because of some valid reasons:
- This is the first paper to provide theoretical analysis of ZSG in the offline RL setting (at least to our knowledge) and establishes a foundation for future work in this area.
- The paper is well-situated within broader RL literature
- The theoretical claims are carefully proven, with reviewer confirming the correctness of major results.

The key enabler for zero-shot generalization in this paper is having task identity labels (i.e., knowing which trajectories come from the same environment or task during training) so we are not doomed to do averages at test time. Yet at the core of the algorithm lies an Oracle for the uncertainty which reduces the applicability. The paper is showing what’s possible when pessimism + structured training are combined, and providing a template that can be extended to more realistic RL settings (and the Oracle is tractable for the linear case).

Despite I regret that the algorithms are not tested (and it would be possible at a reasonable cost on toy tasks such as MDPs or different levels of a simple game), the work introduces an elegant new idea and I think is worth some attention.